# Cell Membrane Fatty Acids and PIPs Modulate the Etiology of Pancreatic Cancer by Regulating AKT

**DOI:** 10.3390/nu17010150

**Published:** 2024-12-31

**Authors:** Carolina Torres, Georgina Mancinelli, Jee-Wei Emily Chen, Jose Cordoba-Chacon, Danielle Pins, Sara Saeed, Ronald McKinney, Karla Castellanos, Giulia Orsi, Megha Singhal, Akshar Patel, Jose Acebedo, Adonis Coleman, Jorge Heneche, Poorna Chandra Rao Yalagala, Papasani V. Subbaiah, Cecilia Leal, Sam Grimaldo, Francisco M. Ortuno, Faraz Bishehsari, Paul J. Grippo

**Affiliations:** 1Department of Biochemistry and Molecular Biology III and Immunology, Faculty of Medicine, University of Granada, 18071 Granada, Spain; 2Instituto de Investigacion Biosanitaria ibs.GRANADA, 18012 Granada, Spain; 3Department of Medicine, University of Illinois Chicago, Chicago, IL 60612, USA; gms891@gmail.com (G.M.); sarah.saeed@advocatehealth.com (S.S.); mckinney@uic.edu (R.M.); akshp64@gmail.com (A.P.);; 4Department of Materials Science & Engineering, University of Illinois Urbana-Champaign, Urbana, IL 61801, USA; emily.chen@lilly.com (J.-W.E.C.);; 5Hospital San Raffaelle, 20132 Milan, Italy; 6Department of Computer Architecture and Computer Technology, University of Granada, 18071 Granada, Spain; 7Department of Medicine, Rush University Medical Center, Chicago, IL 60612, USA; 8Department of Medicine, Division of Gastroenterology and Hepatology, University of Illinois Chicago, 840 S. Wood Street, CSB 708, Chicago, IL 60612, USA

**Keywords:** PDAC, polyunsaturated fatty acids, prevention, PI3K/AKT, PIP2 (phosphatidylinositol 4,5-bisphosphate), PIP3 (phosphatidylinositol (3,4,5)-trisphosphate), diet

## Abstract

**Background:** Pancreatic ductal adenocarcinoma (PDAC) is one of the worst solid malignancies in regard to outcomes and metabolic dysfunction leading to cachexia. It is alarming that PDAC incidence rates continue to increase and warrant the need for innovative approaches to combat this disease. Due to its relatively slow progression (10–20 years), prevention strategies represent an effective means to improve outcomes. One of the risk factors for many cancers and for pancreatic cancer in particular is diet. Hence, our objective is to understand how a diet rich in ω3 and ω6 polyunsaturated fatty acids affects the progression of this disease. **Methods:** We investigated polyunsaturated fatty acid (PUFA) effects on disease progression employing both *in vitro* (PDAC cell lines) and *in vivo* (EL-Kras and KC mice) approaches. Also, we gathered data from the National Health and Nutrition Examination Survey (NHANES) and the National Cancer Institute (NCI) from 1999 to 2017 for a retrospective observational study. **Results:** The consumption of PUFAs in a patient population correlates with increased PDAC incidence, particularly when the ω3 intake increases to a lesser extent than ω6. Our data demonstrate dietary PUFAs can be incorporated into plasma membrane lipids affecting PI3K/AKT signaling and support the emergence of membrane-targeted therapies. Moreover, we show that the phospholipid composition of a lipid nanoparticle (LNP) can impact the cell membrane integrity and, ultimately, cell viability after administration of these LNPs. **Conclusions:** Cancer prevention is impactful particularly for those with very poor prognosis, including pancreatic cancer. Our results point to the importance of dietary intervention in this disease when detected early and the potential to improve the antiproliferative effect of drug efficacy when combined with these regimens in later stages of pancreatic cancer.

## 1. Introduction

Despite a low incidence of just over 1 in 10,000, pancreatic ductal adenocarcinoma (PDAC) remains one of the most aggressive solid malignancies, with a 12% 5-year survival [1] compared to a 68% combined rate for all other cancers. PDAC has been projected to become the second cause of cancer-related deaths in the next few years [2], which is mainly due to two factors: chemoresistance and late diagnosis [3]. Risk factors include family history, age, smoking, obesity, long-standing diabetes, chronic pancreatitis, and diet [4], and recent evidence from genomic sequencing suggests a 15-year interval from initiation to aggressive disease. Both of these suggest a sufficient window of time for early detection [5] and cancer interception [6].

Though more research is needed for earlier detection, there is growing evidence that some cancers, PDAC among them, are influenced by diet and environmental factors [7]. Indeed, 30% of PDAC cases could be prevented by acting on modifiable risk factors (smoking, obesity, alcohol, and/or diet) [8]. Dietary fats, particularly their primary components—fatty acids—play a role in cancer development. However, the relationship between different fatty acid types, including saturated (SFAs), monounsaturated (MUFAs), and polyunsaturated (PUFAs), and PDAC risk remains a topic of ongoing debate and scientific discussion, and the molecular mechanisms exploited by pancreatic cells to store and utilize fat to promote tumorigenesis are not well defined [9]. This has been studied in animal models to assess the high-fat diet (HFD) modification of mutant Kras-induced pancreatic neoplasia and cancer. This includes our seminal work comparing ω3 [10] and ω6 [11] PUFA-enriched diets with standard chow in EL-Kras^G12D^ mice [12] and a similar approach also with docosahexaenoic acid (DHA) and linoleic acid (LA) treatment in human cells [13]. This study revealed a robust reduction in pAKT signaling and associated tumor suppression with ω3 diets, in contrast to the tumor-promoting effects of ω6 diets, highlighting their distinct contributions to pancreatic carcinogenesis. Additional research demonstrates that HFDs (predominantly animal fat/lard) activate Kras and Cox2 in acinar cells [14], reduce FGF21 [15], and expand aerobic glycolysis to generate aggressive PDAC. Work with syngeneic orthotopic models generated similar findings [16], where a HFD (lard) increased cancer growth and fibrosis via CCKR signaling [17].

Previous work described that phospholipid synthesis by tumor cells can be modified by dietary PUFAs [18] and that ω3 PUFA incorporation into phospholipids affects plasma membrane biophysical properties and alters the recruitment/activation of signaling proteins [19] and lipid rafts and their downstream signaling cascades [20]. In a mouse model of colon cancer, ω3 HFD attenuated oncogenic Ras signaling and hyperproliferation by altering its nanoscale proteolipid interactions at the plasma membrane [21].

Yet, few have focused on HFDs enriched in PUFAs employing p48-Cre/LSL-Kras^G12D^(KC) mice [22], which well recapitulate Pancreatic Intraepithelial Neoplasia (PanIN) disease. Previous findings demonstrated that KC mice fed a HFD (corn oil-based; 40% Kcal) promote more advanced pancreatic neoplasia [23], with increased incidence of PDAC, inflammation and dysregulated autophagy [24], peripancreatic visceral adiposity [25], and severe NK cell deficiencies.

To better understand the influence and mechanism of HFDs enriched in PUFAs on PDAC prevention, we investigated PUFA effects on disease progression, employing both *in vitro* (PDAC cell lines) and *in vivo* (EL-Kras and KC mice) approaches. We evaluated the effects of HFDs supplemented with ω3 or ω6 PUFAs on lesion development. Our findings support that ω3-enriched HFDs lower lesion penetrance and cell proliferation associated with reduced pAKT, whereas a diet enriched in ω6-PUFA accelerated tumor formation and proliferation. Using PDAC cell lines incubated with DHA, we observed reduced PI3K/AKT activity. This effect was reversed by linoleic acid (LA) and ω6-enriched HFDs, which can enhance pAKT levels. AkT activation is tightly regulated by its interaction with PIP3 at the membrane. PIP3, once generated by the phosphorylation of PIP2 by PI3K, recruits Akt to the membrane, where it undergoes phosphorylation and becomes active. This activation is crucial for promoting cellular processes such as proliferation, migration, and survival. Conversely, the enzyme PTEN dephosphorylates PIP3 to PIP2, releasing AkT from the membrane and deactivating the pathway. An imbalance in the membrane lipid composition, potentially influenced by dietary fatty acids, can disrupt this regulation [26].

In physiological systems, fatty acids are predominantly transported through the bloodstream bound to serum albumin, which acts as their primary carrier, facilitating their solubility and delivery to tissues for metabolism, storage, or incorporation into biological membranes [27]. To enhance the bioavailability and solubility and reduce degradation, FAs can be incorporated into lipid-based nanoparticles (LNPs) [28]. We used a formulation based on cholesterol and glycerol monooleate, as it has been demonstrated to be stable and have a high fusogenic affinity toward target membranes [29]. We have engineered FA-enriched LNPs, and similar results regarding the inhibition of pAKT, a major downstream effector of Kras, were observed. This is encouraging, because activating KRAS mutations are found in nearly 90% of PDAC cases [30], and it is considered an undruggable target, though the KRAS^G12C^ inhibitor Sotorasib has FDA approval (ClinicalTrials.gov numbers NCT04303780, NCT04185883, and NCT03600883) for adult subjects with advanced KRAS^G12C^ mutant-expressing solid tumors. Yet, most prevalent in PDAC is KRAS^G12D^, which still has no effective inhibitors. Thus, targeting PI3K/AKT serves as a logical alternative.

Although discovery of early biomarkers and detection approaches are needed against PDAC, preventive strategies are equally required to reduce PDAC incidence and mortality. Building on our and others’ previous findings, in this work, we delve deeper into the molecular mechanisms underlying the observed effects of omega-3 fatty acids (DHA). Specifically, we propose a molecular mechanism responsible for the decreased proliferative capacity of pancreatic cells in the presence of DHA. Moreover, we develop lipid nanoparticles (LNPs) as a novel method to enhance PUFA delivery to the cell membrane. These LNPs demonstrated superior efficacy in reducing pAKT levels with ω3 PUFA compared to other standard carriers, such as bovine serum albumin (BSA), underscoring the potential of leveraging biomembrane-based nanostructures to modulate cell signaling. The high biocompatibility and structural similarity of LNPs to cell membranes suggest promising applications for clinical translation in improving drug delivery and therapeutic efficacy. This study aims to not only expand our understanding of the distinct roles of dietary PUFAs in pancreatic cancer progression but also to explore innovative strategies for modulating these pathways, such as the use of lipid nanoparticles for potential clinical translation.

## 2. Material and Methods

### 2.1. Observational Study

GLOBOCAN estimates of cancer incidence and mortality produced by the International Agency for Research on Cancer were used to examine the global burden of cancer attributable to a high body mass index [31].

Data collected from the National Health and Nutrition Examination Survey (NHANES) and the National Cancer Institute (NCI) from 1999 to 2017 were utilized for this retrospective observational study. The NHANES interview included demographic, socioeconomic, dietary, and health-related questions [32]. The examination component consisted of medical, dental, and physiological measurements, as well as laboratory tests administered by highly trained medical personnel. The NCI collected and published cancer incidence and survival data from population-based cancer registries covering approximately 48 percent of the U.S. population. From the NHANES data, the grams of ω3 (*Octadecatrienoic*, *Octadecatetraenoic*, *Eicosapentaenoic*, *Docosapentaenoic*, and *Docosahexaenoic*) and ω6 (*Octadecadienoic* and *Eicosatetraenoic*) PUFAs that each participant consumed were summed and then averaged per year. To account for disease progression and detection, PDAC incidence was lagged 2 years after the PUFA consumption. We investigated the relationship between the increase in the consumption of ω6 and ω3 PUFAs with the increase in PDAC incidence (expressed in %).

According to federal regulations governing research involving human subjects (45 CFR Part 46), Institutional Review Board (IRB) approval is not required for research utilizing publicly available datasets provided that the datasets are publicly accessible and the data are de-identified, uncoded, and stripped of all identifiers. Given that our study adheres to these criteria, IRB approval was not applicable or sought for this research.

### 2.2. Animals and Diets

p48-Cre x LSL-KRAS^G12D^ (KC) and Elastase-KRAS^G12D^ (EL-KRAS) were generated in a C57BL/6 background at an approximate 50:50 male-to-female ratio. Starting at one month of age, animals were provided either a standard diet (SD) or a high-fat diet (HFD) enriched either in ω3 or ω6 PUFAs (for a detailed composition of these diets, see Appendix A) for 8 months and then sacrificed. For euthanasia, mice were anesthetized using ketamine/xylazine (100/10 mg/kg) until unresponsive to toe tap and/or agonal breathing; after which, blood was collected using cardiac puncture. Thoracotomy served as the primary form of euthanasia and exsanguination the secondary form. For organ collection, the pancreas was removed first, and this tissue was divided into three parts. One-third of the pancreas was embedded in a cassette for histological analyses (along with other tissues such as liver, duodenum, lungs, spleen, etc.), another third was stored in RNA later for future RNA studies, and the last third was snap-frozen at −80 °C for protein analyses. Part of the liver was also snap-frozen and stored.

Animals of the genotypes in question were randomly selected (*n* = 5). After being evaluated, no animals of the desired genotype were excluded from any group, and no further randomization was used. All experiments involving the use of mice were performed following protocols approved by the Institutional Animal Care and Use Committee at the University of Illinois at Chicago (UIC), Approval Code: 14-138, Approval Date: 28 August 2014.

### 2.3. Cell Lines, Cell Cultures Conditions, and Treatments

The human PDAC cell lines Panc-1, MiaPaCA-2, and AsPC-1 were obtained from the American Type Cell Culture (ATCC, Manassas, VA, USA). Cells were grown in Dulbecco’s modified Eagle’s medium (DMEM, Sigma-Aldrich) supplemented with 10% FBS (PAA Laboratories, The cell culture company), 20 mM HEPES, 100 U/mL penicillin G, and 100 μg/mL streptomycin (Sigma-Aldrich). All the cultured cells were maintained in a humidified 5% CO_2_ atmosphere at 37 °C. All the experiments were conducted with cells in the exponential growth phase. Cells were tested for mycoplasma contamination every month and annually submitted for lineage testing to assure against any genetic and/or phenotypic drift.

Docosahexaenoic acid (DHA, 22:6 ω3) and linoleic acid (LA, 18:2 ω6) were purchased from Cayman Chemical. To mimic how FAs are transported in the blood, coupled to albumin, both FAs were conjugated with FA-free bovine serum albumin (Sigma-Aldrich, 8.1 μM) for 2 h at 37 °C, as previously done [33], and administered to the cells in 1%FBS DMEM to minimize the lipid component in the FBS. The final concentration of EtOH never exceeded 1% (*v*/*v*). No effect from the exclusive administration of EtOH was observed.

For the exogenous supplementation of phosphoinositide, PIP2 (phosphatidylinositol (4,5)-bisphosphate and PIP3 (phosphatidylinositol (3,4,5)-trisphosphate) were purchased from Echelon Biosciences Inc. and administered to the cells (at 1 µM) also conjugated with BSA in the same conditions described above.

Gemcitabine was kindly provided by UI Health Hospital, dissolved in PBS, filtered, and stored at −20 °C.

### 2.4. Stable Transfections and Transfection Efficiencies of the Panc-1 Cell Line

The GFP-C1-PLCdelta-PH vector (encodes a PIP2 lipid-selective PH domain that can be used as a fluorescent translocation biosensor to monitor changes or local differences in the concentration of plasma membrane PIP2 lipids) was a gift from Tobias Meyer (Addgene plasmid #21179). Also, PH-Btk-GFP (biosensor for PIP3) was a gift from Tamas Balla (Addgene plasmid #51463). The Panc-1 cell line was transfected with these plasmids using Lipofectamine 3000 (according to manufactory recommendations, Thermo Fisher). Cells were selected with G418 (500 μg/mL).

### 2.5. Cell Viability

The viability of the cells was determined by the standard 3-(4,5-Dimethylthiazole-2-yl)-2,5-diphenyltetrazolium bromide (MTT) assay. Briefly, cells were plated in 96-well plates at a density of 5 × 10^4^ cells/mL. They were left overnight to attach to the plate and then incubated with increasing concentrations of DHA (ω3) and LA (ω6). After 48 h incubation, 10 μL of MTT solution (5 mg/mL in PBS; Sigma-Aldrich) was added to the wells and incubated for 3 h at 37 °C. Subsequently, the formazan crystals were solubilized by adding 200 μL of DMSO, and the absorbance was measured at 570 nm.

### 2.6. Immunocytochemistry

Human PDAC cell lines were seeded in multi-well chamber slides at a density of 1.5 × 10^4^ cells/well. After 48 h of incubation with or without the PUFAs, cells were washed with PBS and fixed for 10 min with 4% paraformaldehyde. Before blocking, cells were permeabilized with 0.01%TritonX in PBS for 10 min at room temperature. The blocking was done for 1 h at room temperature using 5%BSA/PBS. Primary antibodies PIP3 and PIP2 were purchased from Echelon Biosciences Inc. (Z-P345) and Santa Cruz (sc-53412), respectively, and diluted to 1:100, then 200 µL/well were added to the chamber slides and incubated overnight at 4 °C. After incubation, the chamber slides were washed three times with PBS and incubated 1 h in darkness and at room temperature with the secondary antibody Alexa Fluor 488 donkey anti-mouse from Life Technologies (1/250), followed by 200 µL of mounting media with DAPI (Cell Signaling). For the transfected cell line, after treatment, cells were fixed, and the GFP signal was assessed.

### 2.7. Histology, Immunohistochemistry (IHC), and Immunofluorescence (IF)

The animal tissues were fixed in 10% formalin (*v*/*v*), paraffin-embedded, and sectioned at 4 μm. The slides were stained with hematoxylin and eosin and with eosin or Masson’s trichrome (Sigma) for histological confirmation of the lesion phenotypes, the surrounding mesenchyme, and their response to the specific diet. For IHC and IF, slides were heated in a pressure cooker using DAKO retrieval buffer (DAKO, Carpinteria, CA, USA). For IHC, endogenous peroxidases were quenched in DAKO peroxidase block for 20 min at room temperature. Tissues were then blocked with 1% BSA (*w*/*v*) in PBS for 1 h at room temperature and exposed to a primary antibody against PCNA and CK19 at 1:100–400 overnight at 4 °C. Slides were visualized using either a HRP-conjugated secondary antibody followed by DAB substrate/buffer (DAKO) or using an Alexa flour 488 or 594-conjugated secondary antibody (Abcam, Cambridge, MA, USA). For PIP3 staining, the purified anti-PtdIns(3,4,5)P3 IgM antibody from Echelon Biosciences was used according to their specific protocol in frozen tissue sections. The immunohistochemical staining was scored by two-blinded investigators (CT and GM). The protein staining level was scored as 0 (no detectable immunostaining), 1 (10–30% immunostaining), 2 (30–60%), and 3 (>60%). The numerical scorings were then averaged.

### 2.8. Western Blot Analysis

Cell or tissue lysates were lysed in RIPA buffer supplemented with protease inhibitors (1 x PBS, 1% Nonidet P-40, 0.5% sodium deoxycholate, 0.1% SDS (*w*/*v*), 0.1 mg/mL PMSF, 0.04 mg/mL aprotinin, and 0.18 mg/mL sodium orthovanadate), followed by homogenization by sonication. Lysates were incubated at 4 °C and cleared by centrifugation at 12,000× *g* for 15 min. The protein concentration was determined by the micro-BCA procedure (Pierce) using BSA as standard. Proteins were denatured in the sample buffer with 5% β-mercaptoethanol (*v*/*v*) at 95 °C for 10 min. Equal amounts of total protein were separated by SDS–PAGE (12%) and transferred to PVDF membranes (GE Healthcare). The membranes were blocked for 1 h at room temperature in TBS-T (10 mM Tris–HCl (pH 7.6), 150 mM NaCl, and 0.05% Tween 20, *v*/*v*) and incubated overnight at 4 °C. The reference protein was β-actin (Santa Cruz, sc-47778). Primary antibodies were detected using a horseradish peroxidase (HRP)-conjugated secondary antibody, also provided by Santa Cruz. The immunocomplex was detected by using the ECL Plus kit (Amersham, Buckinghamshire, UK), and the band density was analyzed with Bio-Rad Imager System software.

### 2.9. GC/MS Analysis of Fatty Acids

In order to assess DHA incorporation into membrane phospholipids, total lipids were extracted from the Panc-1 cell line treated with increasing concentrations of DHA (0–40 µM) with 2:1 (*v*/*v*) chloroform:methanol, as previously described [34]. Total lipids were subsequently separated by thin-layer chromatography with the solvent system of chloroform–methanol–acetic acid–water 90:8:1:0.8 (by vol). Spots corresponding to PC, PE, and TAG were scraped, and FA methyl esters (FAMEs) were prepared by transesterification using methanolic HCl [35]. The FAMEs were analyzed by capillary gas chromatography-mass spectrometry (GC/MS), as described previously [35,36], employing a Shimadzu QP2010 SE equipped with a Supelco Omegawax column. The total ion current in the range of 50–400 *m*/*z* was used for quantification of the FAMEs.

### 2.10. Lipid Nanoparticles (LNPs) Preparation

LNPs were prepared by standard methods based on nanoprecipitation known to yield a uniform distribution of size, shape, and encapsulation efficiency [29,37]. Briefly, LNPs were prepared in NanoAssemblr Ignite (Precision NanoSystems, NanoAssemblr™ Ignite™ (Cytiva) Wilmington, DE, USA) by microfluidics. The desired ratio of lipids (with/without PUFAs) was dissolved in chloroform. After full removal of the organic solvent overnight under a N2 flow, the lipids were then dissolved in ethanol at a concentration of 10 mM. The total flow rate was maintained at 12 mL per min and a 4:1 ratio of aqueous to ethanol inlets. After the removal of ethanol under dialysis for 24 h, the resulting formulation of LNPs dispersed in aqueous buffer was 90% glyceryl monooleate (Sigma-Aldrich, Sigma-Aldrich St. Louis, MO, USA) and 10% cholesterol (Sigma-Aldrich) with 40 μM of DHA or LA. The particles were then added to culture media to stabilize at 37 °C overnight prior to cell treatment. DiO (Invitrogen) for lipid tracing and FA dye were added along with the DHA treatment and used for confocal imaging (LSM800, Zeiss, Zeiss Laboratories, Thornwood, NY, USA).

### 2.11. Statistical Analysis

Values are expressed as the mean ± SD (standard deviation). Differences between two groups were assessed using unpaired two-tailed *t*-tests. The number of biological replicates in all experiments was n ≥ 3. Analyses were performed using GraphPad Prism software v.7.00 (GraphPad Software, Inc., La Jolla, CA, USA). The significant differences were indicated by * *p* < 0.05; ** *p* < 0.005; *** *p* < 0.0005. 

## 3. Results

### 3.1. PDAC Incidence Correlates with ω6 PUFA Consumption

PDAC incidence and mortality rates were derived from GLOBOCAN to showcase the worldwide burden and dietary risk factors. The Human Development Index (HDI) was selected as an indicator of the quality of life (QoL) or development [38]. In relation to the incidence of PDAC, a positive correlation between socioeconomic development and this disease has been previously described [39]. Figure 1A,B show the comparison between incidence and mortality in high and low HDI countries, depending on the cancer site. For PDAC, the estimated number of incident cases in high/very high HDI countries is 462,241, with only 33,291 in medium/low HDI countries, representing a nearly 14 times higher rate of PDAC cases in developed countries. This is consistent with the average body mass index (BMI), which increases more rapidly because of societal and economic changes. We next evaluated the burden of PDAC associated with obesity. Figure 1C,D show cancer incidences among females (C) and males (D) worldwide attributable to excess BMI. In total, 14.5% of all PDAC cases demonstrate a causal link with obesity.

Since obesity is a causal risk factor, we explored how HFD (particularly enriched in ω3 and ω6 PUFAs) contributed to PDAC incidence from 1999 to 2017. Data were collected from the National Health and Nutrition Examination Survey (NHANES) and the National Cancer Institute (NCI). The time from the start of recruitment to establishing the PDAC incidence rate was 2 years. Figure 1E compares increases in PDAC incidence with increases in consumption of ω3 and ω6 PUFAs and indicates that smaller increases (or even decreases) in PDAC incidence were associated with a greater increase in ω3 over that in ω6 consumption. In 2001, the increase in ω3 consumption was five times higher than the increase in ω6 consumption (~7.6% vs. 1.5%), and the increase in PDAC incidence was low (2%). In 2007, the increase in ω3 exceeded the increase in ω6, and the increase in incidence was only 0.8%. In 2011, the increase in ω3 was 1.3 times higher than the increase in ω6 (13.12% vs. 10.1%), leading to a 2.4% reduction in PDAC incidence. Finally, in 2015, the increase in ω3 consumption was 3.86 times that of the ω6 consumption (0.93% vs. 0.24%), though the incidence remained unchanged, likely due to the relatively low level of increase (less than 1%). On the contrary, in 2005, the higher increase in incidence (7.8%) occurred when the lowest ω3 consumption was registered (−3.75%), and ω6 consumption was 3.4 times higher than that of ω3. These data are available in the Appendix A.

This observational data thus indicate that changes in fat consumption correlate with PDAC rates, especially when there are stark differences between ω3 and ω6 consumption, suggesting a population-level link between imbalanced ω6 over ω3 intakes and higher rates of the disease over time. To further support this notion, we administered ω3 and ω6 FAs to human PDAC cells in culture and in the diets of mice that developed preinvasive lesions.

### 3.2. Ω3 PUFA-Enriched Diet Protects Against Neoplastic Lesion Formation in KRAS^G12D^ Mice

To investigate the influence of PUFAs in the early development of PDAC, both EL-Kras (EK; model of cystic papillary lesions similar to IPMN) and p48-Cre; LSL-KRAS^G12D^ (KC; model of PanIN lesions) animal models were fed for 8–9 months with the standard diet (SD), ω3 (menhaden oil)-, or ω6 (safflower oil)-enriched HFDs. The general composition of the diets and the specific lipid profiles can be found in Appendix A. The animals were sacrificed, and the tissues were fixed and analyzed. We evaluated tumor progression by means of the percent of normal tissue and total number of lesions per field of view for KC (Figure 2A) and EK mice (Appendix A). Compared to KC mice fed SD, ω3-fed mice had a 40% reduction in PanIN frequency (Figure 2A,D), 39% less fibrosis (Figure 2B,E), and 68% reduced cell proliferation/PCNA staining in CK19-positive neoplastic tissues (Figure 2C,F).

Interestingly, the occurrence of neoplastic lesions was significantly accelerated by the ω6 diet, compared both to SD and especially to ω3. Ω6-fed animals presented with a 35.7% higher frequency of PanIN lesions compared to the SD-fed animals and 125% higher than the ω3-fed animals (Figure 2A,D). Regarding fibrosis, ω6-fed mice showed a 22.8% increase in collagen staining compared to SD-fed mice and a 100% increase compared to the ω3-fed animals (Figure 2B,E). PCNA staining in CK19-positive neoplastic tissue was likewise induced, with a 36.7% higher staining in ω6-fed animals compared to those fed with SD and an over three times higher proliferation rate than the ω3-fed animals (Figure 2C,F). Very similar results were obtained in the EK mouse model (Appendix A). These data suggest ω3 and ω6 have opposing roles in neoplastic development *in vivo*.

### 3.3. PUFAs Modulate the KRAS Downstream Effector PI3K/AKT

To investigate the molecular mechanisms governing the reduction in neoplastic lesions, RAS downstream MAPK/ERK and PI3K/AKT pathways were evaluated. Neither of the diets had an effect over the MAPK/ERK pathway (Figure 2G,H). The ω3-enriched diet induced a significant 92% decrease in the pAKT/AKT levels in KC mice (Figure 2G,H). Downregulation of pAKT was consistent with reduction of the target proteins controlled by pAKT, such as FOXO3a and BAD (Figure 2G,H), which, when phosphorylated, promote cell survival, growth, proliferation, and inhibition of cell death [40], respectively. The levels of pFOXO3a and pBAD were undetectable in ω3-fed KC mice, exhibiting a 90% reduction compared to SD controls (Figure 2G,H). Yet, the diet enriched in ω6 resulted in a significant 124% and 2700% increase in the pAKT/AKT ratio in ω6 compared to SD and ω3, respectively. The ω6 diet significantly promoted the phosphorylation of FOXO3a and BAD (Figure 2G,H). All these results were also similarly observed in the EK model (Appendix A) and suggest neoplasia in mice fed ω3 and ω6 PUFA-enriched diets results in opposing regulation of the AKT pathway.

### 3.4. Exogenous Administration of Fatty Acids Affects the Viability of PDAC Cell Lines, Modifying Signaling Through PI3K/AKT Pathway

To study the role of PUFAs on cell viability, human PDAC cell lines (Panc-1, MiaPaca-2, and AsPC-1) were treated with DHA and LA. These fatty acids were chosen because they were the predominant fatty acids in the respective diets (Appendix A), and we further confirmed by GC/MS that they were indeed the lipid species that were preferentially incorporated into the tissues of the animals (Appendix A). After 48 h treatment, cell viability and cytotoxicity were assessed by the MTT assay. DHA was able to induce PDAC cell cytotoxicity in a dose-dependent manner in these three cell lines, with an IC_50_ of 40 µM (Figure 3A–C). No significant effect was detected over cell growth when the cells were incubated with LA though at low concentrations, LA was able to promote cell viability (Figure 3D–F). These data suggest that ω3 and ω6 have inverse roles in sustaining PDAC cells, since the former reduce cell viability while the latter either do not affect or induce it.

To understand the importance of the ratio between ω3 and ω6 PUFAs rather than their individual roles, we next determined the impact of a combination of these two fatty acids. The recommended dietary ω3:ω6 ratio is between 1:1 and 1:4. We incubated the pancreatic cancer cell line Panc-1 with 40 µM of both fats. Our results show that the presence of DHA is sufficient to palliate the induction of proliferation exerted by LA (Figure 3G). Indeed, we also analyzed the effect of other possible combinations of these fatty acids with other ratios, and the results are similar (Appendix A). One interesting caveat is that the amount of DHA seems to be more critical than the actual ω3:ω6 ratio. However, regardless of the amount and/or ratio, we see a prevalence of the protective effect of DHA even when LA is also present in the medium.

Additionally, we examined whether the effect of PUFAs could condition other factors that affect the PDAC cell proliferation, including gemcitabine, a widely used drug in PDAC therapy. Panc-1 cells were incubated in increasing concentrations of gemcitabine, alone or in combination with DHA and LA. Our data indicate that, while gemcitabine inhibits Panc-1 cell proliferation at high concentrations (IC_50_ = 50 µM), 5 µM of DHA achieves the same effect at an eight-times lower gemcitabine concentration. In contrast, 5 µM of LA in culture media requires twice the concentration of gemcitabine to reach the same IC_50_ (Figure 3H).

To establish that ω3 and ω6 PUFAs regulate the AKT pathway similar to that observed *in vivo*, Panc-1, MiaPaca-2, and AsPC-1 cells were treated with a concentration gradient 0–40 µM of DHA for 48 h, and AKT pathway signaling was assessed by immunoblotting. All the three cell lines showed a significant reduction in AKT activation (expressed as the pAKT/AKT ratio) at 40 µM (80, 84, and 80% reduction for Panc-1, MiaPaca-2, and AsPC-1, respectively) when compared to the untreated control (Figure 4A–C,G–I, green bars). Significant suppression of AKT activation was already achieved at lower concentrations: 10 µM for Panc-1 and AsPC-1 (67 and 46% reduction, respectively) and 20 µM for MiaPaca-2 (38% reduction). We also investigated the effect of a LA gradient concentration 0–40 uM on AKT phosphorylation in these cell lines. Conversely, LA significantly induced signaling through this pathway at 40 µM in all cell lines compared to the untreated control (30, 60, and 12% increase for Panc-1, MiaPaca-2, and AsPC-1, respectively; Figure 4D–I, red bars). MiaPaca-2 cells were able to significantly induce AKT activation at lower LA concentrations (20% increase at 10 and 20 µM). These data suggest that DHA and LA regulate AKT phosphorylation in human PDAC cells lines like that in mouse pancreatic neoplasms.

To gain insight into the molecular alterations induced by these PUFAs, Panc-1, MiaPaca-2, and AsPC-1 cells were then treated with 40 μM of DHA or LA for 48 h. Regulators of AKT phosphorylation were then analyzed by immunoblotting. Western blot analysis revealed no significant changes in the PDK1, PTEN, or PI3K protein expression levels (Figure 4J–L). However, the effects on the downstream effector of pAKT, pBAD (and BAD), were the same as those obtained in the animal samples (Figure 2G and Figure 4J–L). These data suggest that PUFAs are not modulating upstream regulators of the AKT pathway.

### 3.5. Exogenous Administration of Fatty Acids Alters the Composition and Localization of Phospholipids in the Membrane

Since PUFAs modulate AKT phosphorylation but not key upstream regulators, we next investigated if there were any changes in DHA/LA recruitment to the membrane. Activation of AKT is initiated at the plasma membrane, where PIP3, generated by PI3 K and dephosphorylated by PTEN, recruits AKT to the membrane [41]. Since diets can modify the membrane FA composition [42], we next sought to assess the DHA uptake (22:6n-3) into the Panc-1 plasma membrane (phospholipids like phosphatidylcholine (PC), phosphatidylethanolamine (PE), and phosphatidylinositol (PI) and triglycerides (TG)) following incubation with 0–40 μM DHA for 48 h. Incorporation was assessed by GC-MS after thin layer chromatography (TLC) separation. All these species but PI were detectable by this technique. We identified that DHA treatment significantly increased the percent of DHA in PC, PE, and TGs in a dose-dependent manner (Appendix A). Based on these data, it appears that, among the various potential effects that dietary PUFAs can have on cellular physiology, in our model, the primary mechanism involves their incorporation into cell membranes. This incorporation seems to play a pivotal role in triggering the observed changes in signaling pathways, particularly in the PI3K-AKT cascade, rather than other pleiotropic effects commonly associated with fatty acids. These findings highlight the significance of membrane remodeling as a central driver of the phenotypic effects seen in pancreatic cancer cells.

A low PI concentration in the membrane prevented measuring its concentration. We considered visualizing the membrane PIP3 by immunofluorescence following PUFA treatment to modify the PIP2/PIP3 ratio and downstream AKT activation. Pancreas tissue from 9-month-old ω3 HFD-fed EK mice showed significant 50% and 71% less membrane PIP3 staining in acinar cells compared to mice fed the SD and ω6 HFD, respectively (Figure 5A,B).

To determine if these results could be recapitulated in human PDAC, Panc-1, MiaPaca-2, and AsPC-1 were incubated with 40 µM DHA or LA for 48 h (Figure 5C–F), which recapitulated our *in vivo* findings. With DHA treatment, there was a significant reduction (38%, 34%, and 27% in Panc-1, MiaPaca-2, and AsPC-1, respectively) in PIP3 staining compared to the control. Conversely, LA treatment showed a significant increase in membrane PIP3 staining (88%, 50%, and 47% and 202%, 128%, and 98% compared to the control and DHA in Panc-1, MiaPaca-2, and AsPC-1, respectively). These data suggest that, regardless of the model examined (*in vitro* or *in vivo*), ω3 and ω6 PUFAs have inverse effects on the levels of PIP3 in cell membranes, since the former significantly reduces staining while the latter increases it. These data are consistent with reduced pAKT in mice and human cells in the presence of the ω3 FAs (Figure 2E,F and Figure 4) when compared to those in the presence of ω6 FAs.

### 3.6. Exogenous Administration of PIPs Is Able to Restore Signaling Through AKT

To confirm DHA and LA modification of the PIP content, we pursued a gain of function analysis with exogenous PIP2 (exoPIP2) and PIP3 (exoPIP3). We previously showed DHA was able to reduce AKT signaling, impacting the PIP2/PIP3 ratio in Panc-1 cells (Figure 5C,D). Hence, we supplemented 40 µM DHA treatment Panc-1 cells with 1 µM exoPIP3 (Echelon Biosciences Inc., Salt Lake City, Utah, USA). As expected, the combined DHA and exoPIP3 resulted in the increased presence of membrane PIP3 and subsequent activation of pAKT, opposing the effect of DHA alone (Figure 5G,H). Conversely, Panc-1 cells treated with LA and supplemented with 1 µM exoPIP2 decreased PIP3 staining and pAKT activation, like those observed in the controls (Figure 5G,H). These results support that DHA promotes a higher PIP2 to PIP3 ratio consistent with the reduction in pAKT and cell proliferation. However, LA demonstrated a lower PIP2-to-PIP3 ratio and subsequent increase in pAKT levels and cell proliferation.

### 3.7. Exogenous Administration of Fatty Acids Alters the Cellular Localization of PH-Btk-GFP Fusion Proteins

To better understand PI dynamics in cells via subcellular localization, we utilized PIP2 and PIP3 GFP-fused biosensors to determine if we could recapitulate these findings. PH-Btk-GFP expresses GFP fused to the BTK plekstrin homology domain, which has specificity in binding PIP3, thus being considered a PIP3 biosensor [43]. Panc-1 cells were transfected with PH-Btk-GFP plasmids and treated with 40 µM of both PUFAs. After 48 h, enriched plasma membrane GFP staining was observed in LA-treated Panc-1 cells (Figure 6A, white arrows, and Figure 6B). A GFP signal could also be detected in both BSA- and DHA-treated transfected cells, verifying the transfection efficiency. However, DHA treatment seemed to prevent PIP3 membrane localization in tumor cells, keeping GFP signals diffused in the cytoplasm. For BSA control cells, some membrane staining was detected but at significantly lower levels than LA-treated cells (Figure 6A,B). These findings demonstrate that DHA prevented the membrane presence of PIP3, which is a docking station for AKT phosphorylation, indicating that DHA can modulate AKT activity by inhibiting PIP3 membrane localization. LA induced AKT activation by promoting PIP3 localization to the membrane.

### 3.8. DHA Incorporated into PIP2 Reduces the Affinity of PI3K

To demonstrate DHA modulation of the membrane PIP3 levels, we investigated if DHA could directly alter the PI3K-PIP2 interaction, where the sn-2 position of PIP2 was typically occupied by an unsaturated FA [44]. PUFA supplementation could induce sn-2 FA moiety change that can impact signaling pathways regulated by membrane lipids.

We transfected Panc-1 cells with the GFP-C1-PLCdelta-PH vector (PIP2 biosensor [45]) to study if the affinity of PI3K to PIP2 is modified by PUFAs. Cells were incubated with 40 µM of DHA or LA for 48h and co-immunoprecipitated to measure the interaction between PI3K and PIP2. The GFP-tagged PIP2 biosensor was pulled down by GFP antibodies, and PI3K was measured by Western blotting (Figure 6C,D) and compared to the total PI3K without pulldown. DHA treatment reduced PI3K interaction with PIP2 but did not alter the total levels of PI3K, in agreement with previous results (Figure 4J). In the presence of DHA, there was a ~50% reduction in PI3K interactions with PIP2 when compared to cells treated with LA. These results support that increased PIP2/PIP3 ratios following DHA treatment and consequent pAKT reduction may be due to modification of the affinity of PI3K for its PIP2 substrate.

### 3.9. Exogenous PUFA Administration by Lipid Nanoparticles (LNPs) Mimics the Findings Obtained with BSA-Conjugated PUFA

To reproduce and extend these results (and improve on albumin to deliver PUFAs), lipid nanoparticles (LNPs) were customized with DHA included in their composition. After incubating Panc-1 cells with DHA-enriched LNPs, the physiological effect was tested by measuring the pAKT levels after 48 h. As observed in Figure 6E, the nanoparticle-based delivery system did not affect AKT expression compared to BSA. As shown in Figure 4, the treatment of PDAC cells with BSA-conjugated DHA significantly reduced AKT activation (69% reduction). LNP delivery recreated the results obtained with BSA and reduced pAKT expression by 87%, indicating that LNP delivery is superior in pAKT reduction (Figure 6E,F). Interestingly, fluorescent staining demonstrated that BSA+DHA-treated cells had FAs throughout the entire cell, while the LNP-DHA-treated cells exhibited FA predominantly in the inner plasma membrane, where these particles were retained (Figure 6G).

## 4. Discussion

PDAC incidence and mortality continue to rise [8,46], despite advancements in the knowledge of its etiology and risk factors. Early detection is paramount and has a profound benefit for patient survival [47] but requires effective preventative measures to interrupt tumor progression [3], which is possible with more than a decade of time between neoplasia and advanced disease [48]. A number of such environmental factors include tobacco exposure, alcohol use, diabetes, chronic pancreatitis, obesity, and diet. With an epidemiological association between PDAC incidence and mortality in developed countries, having a prevalence for HFDs is likely a starting point for dietary interventions [8]. Indeed, there is a decreased risk with the consumption of fruit, vegetables, and folate-rich foods [49,50] and increased risk associated with the intake of red and processed meat [51]. These findings have been well tested in several PDAC mouse models. There has been a prominent turn from animal fat to a higher intake of ω6 PUFAs due to their cardiovascular benefit. Yet, it is important to assess the type of PUFA in other diseases, including cancer. The Healthy Lifestyle Index (HLI) combines information on smoking, alcohol intake, dietary habits, BMI, and physical activity and was related to PDAC within the EPIC study [52], which showed that adherence to a combination of healthy lifestyle habits was strongly inversely associated with PDAC risk. Epidemiological studies are crucial to learn population behavior and disease patterns, particularly for relative low incidence pathologies such as pancreatic cancer. Several epidemiological studies associate increased PDAC incidence with dietary fat and obesity [53,54,55,56]. The biological basis underlying this association still requires further study, with a specific focus on the role of each individual dietary fat. Our findings align with the broader understanding of PDAC as a multifactorial disease influenced by lifestyle and dietary factors.

A hallmark of cancer cells, including PDAC, is their ability to sustain unlimited proliferation, which requires substantial metabolic and structural resources. Fatty acids, particularly PUFAs, serve not only as membrane components but also as substrates for energy metabolism and mediators of cell signaling [57]. Thus, lipogenesis is accelerated in cancer cells, as high proliferative rates lead to preferential fatty acids uptake from the environment [58]. PDAC cells have been shown to utilize exogenous fatty acids and stored lipids to enhance their metastatic capacity [59].

Socioeconomic changes likely contribute to the change in the proportion of fats ingested in the diet. Changing dietary habits have led to a change in fatty acid consumption, with an increase in ω6 fatty acids and a marked reduction in the consumption of ω3 fatty acids. In addition, the ω6 FAs content increased considerably as a result of the rise (up to 70%) in ω6-rich grains and also the addition of vegetable oil. This, in turn, has resulted in an imbalance in the ω6/ω3 ratio, very different from the original 1:1 ratio that humans had in the past [60]. In this context, this imbalance has been highlighted as a factor contributing to disease development and progression. Both the analysis of data from the NHANES population database and findings from a recent study consistently indicate that a higher ω6/ω3 ratio is associated with an increased risk of certain health conditions [61]. These results underline the importance of dietary balance between these fatty acids and provide further evidence for the role of ω3 and ω6 polyunsaturated fatty acids in disease development and prevention.

Our study contributes to this field by demonstrating that ω3 and ω6 PUFAs play a central role in PDAC progression in our model, primarily through their modulation of AKT signaling pathways. While PUFAs are known to influence various cellular processes, our findings suggest that their impact on Akt signaling is the predominant mechanism affecting tumor cell proliferation and survival in this context [13,62,63]. AKT is a serine-threonine protein kinase with roles in cell growth, proliferation, and apoptosis in a PI3K-dependant manner. PI3K, its substrate (PIP2), and product (PIP3) are considered obligate and rate-limiting for proper AKT activation. Inactive cytosolic AKT is recruited to the membrane and engages PIP3 through AKT PH domain binding, leading to phosphorylation and the activation of AKT (reviewed in [64]). In this work, we focused on how PUFAs modify these functional upstream modulators of AKT by altering the configuration of membrane phospholipids and subsequent changes in pAKT, leading to the promotion or inhibition of cell proliferation.

This preventive effect was confirmed in two mouse models with different degrees of lesion histotypes (cystic papillary neoplasms or CPNs and PanINs) and kinetics. The ω3 HFD was able to reduce the progression and development of neoplastic lesions in these animals with a substantial reduction in lesion frequency, cell proliferation, and fibrosis. Under the condition of excess ω6 FA-enriched HFDs, a prevalence of neoplastic lesions was identified: a substantial increase in frequency with a higher proliferative index and more fibrotic tissue. Although the difference between ω3 HFD and SD was significant, a more dramatic change was obtained between the two HFD groups. Other studies based on ω3 HFD in animal models of carcinoma have described a reduction in tumor size and improved survival [65,66]. Fat-1-p48^Cre/+^-LSL-KRAS^G12D/+^ exhibited a dramatic inhibition of PDAC incidence, frequency of PanIN-3 formation, and their progression to PDAC [67]. Fat-1 mice convert ω6 to ω3 FAs, which can mitigate potential confounding dietary factors. Orthotopic PDAC mice fed a HFD (enriched in oleic and linoleic acids) had increased tumor growth and metastasis [16]. Our results suggest that DHA-enriched diets mitigate neoplastic progression in mouse models, supporting the potential of ω3 PUFAs as co-adjuvants in cancer therapy.

The incubation of cancer cell lines with DHA demonstrated multiple anticancer mechanisms of action, including inhibition of cell proliferation [68,69], migration, invasion [70], and apoptosis-inducing capacity [20,71,72]. Our *in vitro* results with PDAC cells confirm an antiproliferative effect of DHA concomitant with a reduction in the activation of AKT-mediated proliferation and subsequent reduction of downstream AKT effectors BAD and FOXO3a. We observed an increase in the total level of BAD, which interacts with either Bcl-2 or Bcl-X_L_, neutralizing their anti-apoptotic functions [73]. DHA also induced the total level of unphosphorylated FOXO3a, which is a tumor suppressor factor. Recently, it has been shown that FOXO3a expression was remarkably reduced in PDAC tissues and correlated with metastasis-associated pathologic characteristics and poor prognosis in PDAC patients [74]. Hence, the inhibition of PDAC cell proliferation can be mediated by DHA induction of FOXO3a, as others have described previously [75]. Interestingly, the protective role of ω3 seems to be more prevalent when both fats are present.

With no significant difference in levels of the regulators of AKT phosphorylation, we aimed to study if PUFA cell membrane incorporation altered AKT-regulated signals. We observed that supplemental DHA influenced the plasma membrane composition, aligning with previous studies that have demonstrated its potential role in modifying phospholipid profiles. These findings support the idea that DHA can affect cellular processes by altering membrane-associated signaling mechanisms, consistent with the established research [76,77,78]. DHA treatment increased, while LA reduced the PIP2:PIP3 ratio in PDAC cell membranes. DHA likely modifies the structure of PIP2, reducing its affinity for PI3K, which is required to convert PIP2 into PIP3 [79]. The dissociation of PI3K from PIP2 prevents the further production of PIP3, which prevents AKT phosphorylation and activation, thereby disabling AKT signaling. The proposed mechanism is depicted in Appendix A.

Since PUFAs appear to be pleiotropic, impacting metabolic and non-metabolic pathways, it is essential to further dissect all these mechanisms to demonstrate a direct means by which diet functions as a preventive strategy or even as a co-adjuvant treatment. ω3 PUFAs have been shown to act synergistically with some chemotherapeutic or chemopreventive agents [69,80,81]. Our results support this idea, since the antiproliferative capacity of gemcitabine *in vitro* was enhanced when administered in combination with DHA, even at lower concentrations. This reduced the dose of gemcitabine necessary to inhibit growth in the cell culture by 50%, suggesting this approach could be amenable to an additional anticancer agent. In a recent study investigating intravenous ω3 PUFAs and gemcitabine chemotherapy vs. gemcitabine therapy only in patients with PDAC, the combined treatment significantly reduced the immune modulatory cells (particularly, myeloid-derived suppressor cells) and stability of Tregs [82].

One area that needs exploration would be related to optimizing the administration of PUFAs to minimize side effects that may occur, such as abdominal pain or diarrhea, and allowing the rapid incorporation into cell membranes. With the idea of improving the delivery method of ω3 PUFA to pancreatic cells, we have developed DHA-containing lipid nanoparticles (LNPs). Our findings indicate that DHA-LNPs can be utilized for PUFA uptake in cells and impact AKT-regulated proliferation. Modification of the lipid profile of liposomes has been shown to directly have an impact on the proliferation of breast cancer cells incubated with them [83]. A low-density lipoprotein (LDL)-based nanoparticle with DHA was engineered to enhance the physical, oxidative stability, and delivery of DHA to target cells, achieving selective cytotoxicity toward hepatocarcinoma cells and enhanced tumor cell death through ferroptosis [84,85]. The application of ω3 PUFA nanoformulations in lung and prostate cancer demonstrated that the simultaneous presence of DHA in the nanoparticles enhanced the anticancer activity of taxanes both *in vitro* and *in vivo* and also increased the mean survival time of mouse models [28]. More research needs to be done, especially in PDAC, including *in vivo* evaluations of nanoformulations in the preclinical setting, especially in combination with one or more chemotherapeutic agent within these LNPs. In addition, phospholipid and fatty acid lipidomics represent valuable future avenues of research to confirm and extend our findings. These approaches could provide deeper insights into the specific lipid species involved in modulating signaling pathways and membrane dynamics, further validating the observed effects of dietary PUFAs on pancreatic cancer progression.

## 5. Conclusions

In conclusion, our work reinforces the importance of dietary PUFAs in modulating cancer cell behavior and suggests a significant role for membrane lipid composition in PDAC progression. Additionally, the development of novel delivery systems, such as DHA-containing lipid nanoparticles (LNPs), offers promising avenues for optimizing the incorporation of PUFAs into cancer therapies. These approaches not only enhance the bioavailability of PUFAs but also maximize their therapeutic potential by targeting specific signaling pathways and minimizing the side effects.

These findings align with the broader goals of precision nutrition and nutraceutical research, underscoring the need for further studies to explore the integration of fatty acid-based interventions into preventive and therapeutic strategies for PDAC. Considering the relatively poor survival of PDAC patients, prevention and/or combination therapeutic strategies are critically needed to reduce the mortality of this aggressive cancer.

## Figures and Tables

**Figure 1 nutrients-17-00150-f001:**
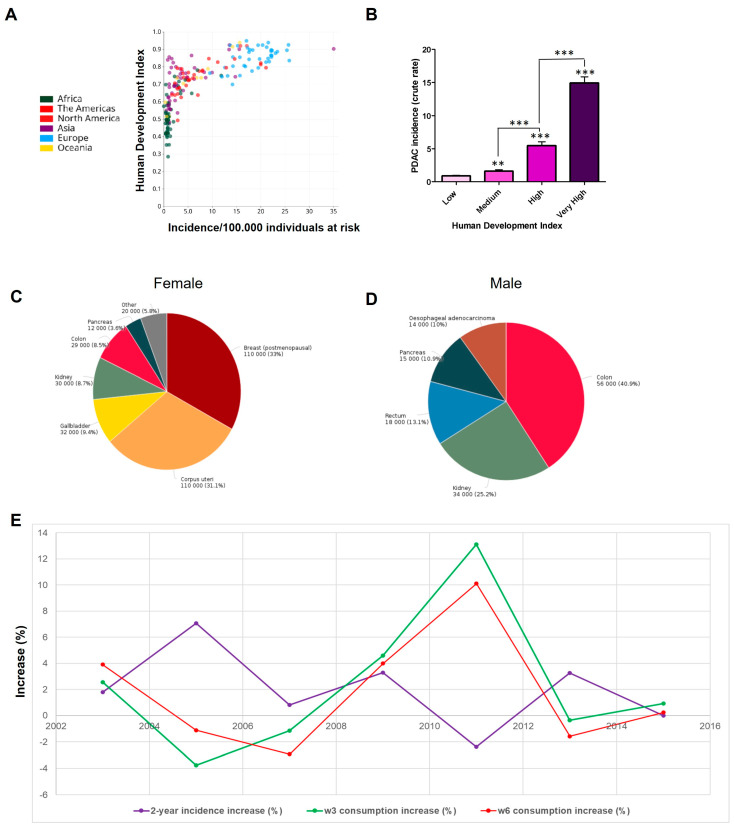
PDAC incidence correlates with ω6 PUFA consumption. (**A**) Pancreatic cancer incidence (crude rate) as a function of the Human Development Index (HDI). For a specific tumor in a given population, crude rates are calculated simply by dividing the number of new cancers or cancer deaths observed during a given time period by the corresponding number of individuals in the population at risk. For cancer, the result is commonly expressed as an annual rate per 100,000 individuals at risk. (**B**) PDAC incidence (crude rate) is rated by the 4-tier Human Development Index (HDI Low (<0.550); Medium (0.550–0.699); High (0.700–0.799); Very high (>0.800)) based on the United Nation’s 2019 Human Development Report. Cancer cases (at all anatomical sites) among females (**C**) and males (**D**) worldwide attributable to excess body mass index are shown by anatomical site as a percentage of the total number of all such attributable cases at all anatomical sites in the studied population. (**E**) Incidence of PDAC in relation to the consumption of PUFAs in the period of time from 1998 to 2018. Grams of both fats consumed were summed and averaged per year. Data show the percentage of increase for both the consumption of PUFAs (2 years before) and incidence. Asterisks in the graphs above a group define significance against the SD control: ** *p* < 0.005; *** *p* < 0.0005. Bars indicate significant differences between two groups. Results are expresses as mean ± standard deviation (SD).

**Figure 2 nutrients-17-00150-f002:**
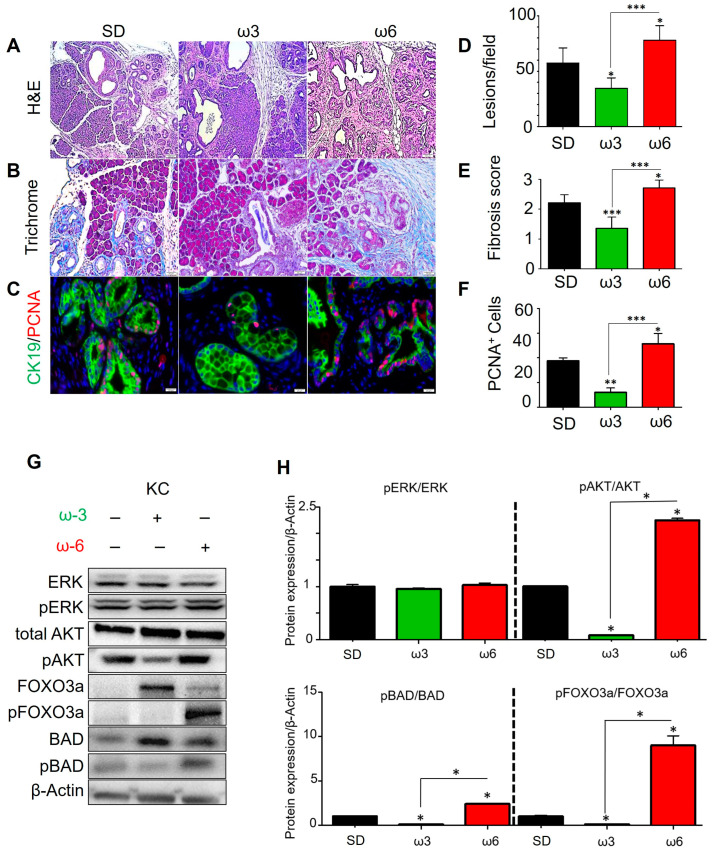
Dietary PUFAs influence the progression of pancreatic neoplasm. KC mice were fed with the standard diet (SD), ω3 (menhaden)-, or ω6 (safflower)-enriched diets (n = 5) for 9 months. (**A**) Representative H&E images of each diet group and the number of lesions were counted per high power field of 5 different fields of view and averaged (10×). (**B**) Representative images of trichrome staining for each diet group and corresponding fibrosis score (5 different fields of view, 20×)). (**C**) Representative images of PCNA and CK-19 double staining with quantitation of the number of positive PCNA nuclei per high power field (5 different fields of view, 40×). (**D**–**F**) The pancreata from KC mice were evaluated by H&E and scored by two independent investigators (CT and GM) for the total number of PanIN lesions (**D**), fibrosis (**E**), and proliferation (**F**), counted per high power field of 5 different fields of view and averaged. (**G**) Western blot images of KC mice pancreata on each diet probing for total and phosphorylated ERK and AKT proteins. Downstream regulators of the AKT pathway were also probed, including total and phosphorylated Foxo3a and BAD proteins. B-actin used as a loading control. (**H**) Averaged quantification of the replicates performed by immunoblotting, relative to the β-actin expression level. The bar graphs represent the ratio of phosphorylated protein to its non-phosphorylated form, which serves as an indicator of activation. Quantification was done using the ImageJ (NIH) software. All the images are representative of the averaged results of the scoring of KC (n = 5). Asterisks in the graphs above a group define significance against the SD control: * *p* < 0.05; ** *p* < 0.005; *** *p* < 0.0005. Bars indicate significant differences between two groups. Results are expressed as the mean ± standard deviation (SD).

**Figure 3 nutrients-17-00150-f003:**
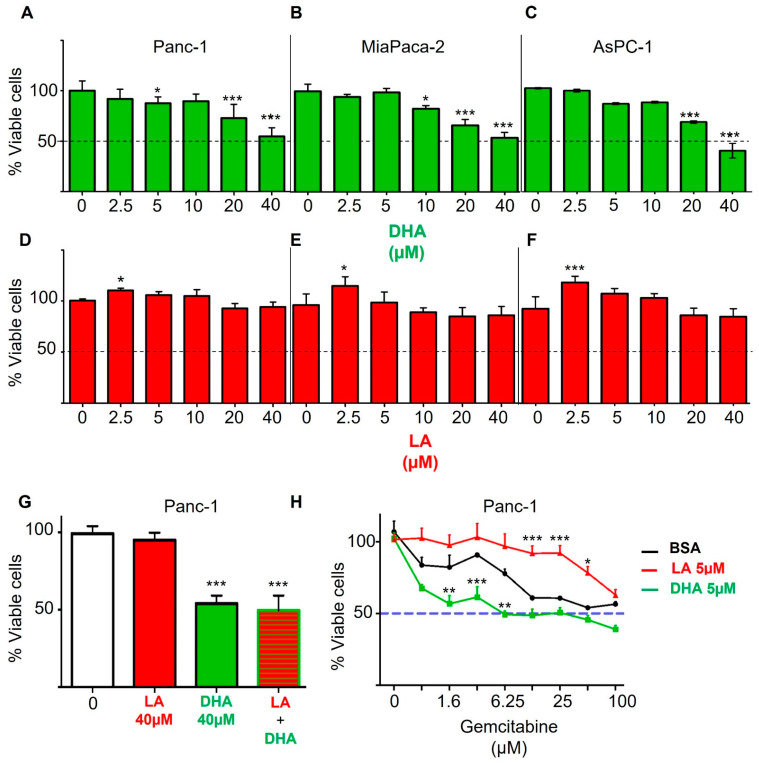
Exogenous supplementation of PUFAs *in vitro* modulates the viability of pancreatic cancer cells. Pancreatic cancer cell lines Panc-1 (**A**), MiaPaca-2 (**B**), and AsPC-1 (**C**) were incubated for 48 h with increasing doses of DHA, and cell viability was determined by MTT assay. Pancreatic cancer cell lines Panc-1 (**D**), MiaPaca-2 (**E**), and AsPC-1 (**F**) were incubated for 48 h with increasing doses of LA, and cell viability was determined by MTT assay. All the assays were performed in triplicate and averaged. (**G**) The Panc-1 cell line was cultured with the same amount of DHA and LA (1:1 ratio) for 48 h, and viability was determined by MTT. (**H**) The Panc-1 cell line was cultured with increasing concentrations of the drug gemcitabine (0–100 µM) with or without the addition of one of the fatty acids (5 µM). The treatment was maintained for 48 h, and the viability was determined by MTT. Asterisks above a group define significance against the corresponding untreated control: * *p* < 0.05; ** *p* < 0.005; *** *p* < 0.0005. Results are expressed as the mean ± standard deviation (SD).

**Figure 4 nutrients-17-00150-f004:**
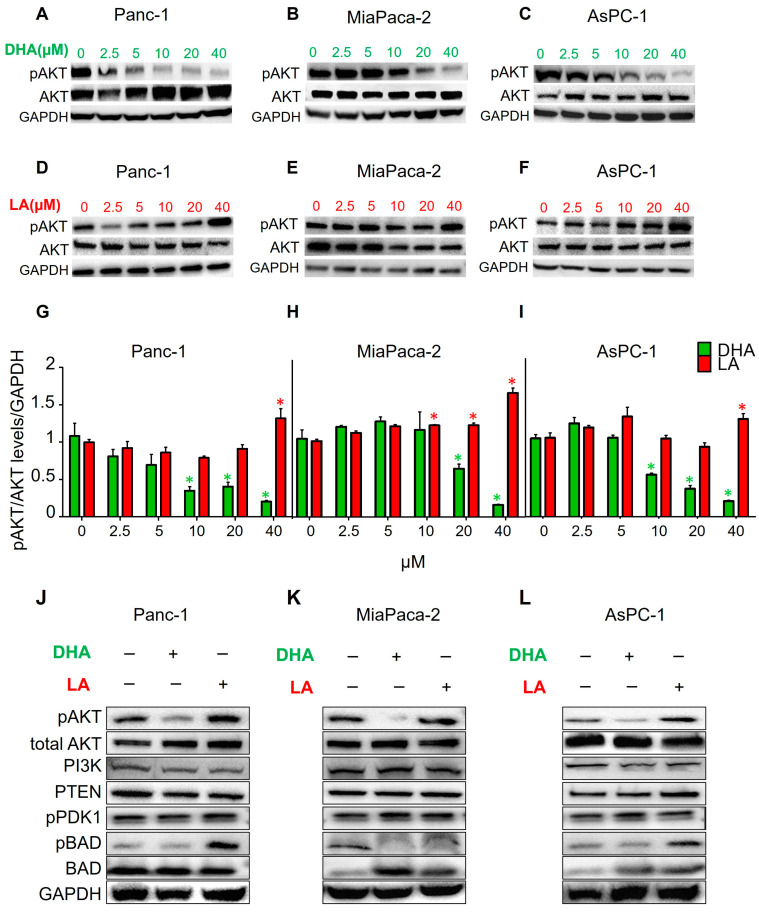
Exogenous supplementation of PUFAs *in vitro* modulates the AKT pathway in pancreatic cancer cells. pAKT/AKT Western blot analysis of Panc-1 (**A**), MiaPaca-2 (**B**), and AsPC-1 (**C**) incubated for 48 h with increasing concentrations of DHA and Panc-1 (**D**), MiaPaca-2 (**E**), and AsPC-1 (**F**) incubated for 48 h with increasing concentrations of LA. (**G**–**I**) Averaged ratio of phosphorylated AKT (pAKT) to total AKT (pAKT/AKT) for each cell line and treatment, representing the level of AKT activation under the specified conditions. Western blot images of Panc-1 (**J**), MiaPaca-2 (**K**), and AsPC-1 (**L**) cells treated with 40 μM of DHA or LA probing for pAKT/AKT and key regulators of AKT, including PI3 K, PTEN, PDK1, and BAD. Quantification of the replicates performed by immunoblotting, relative to the GAPDH expression level. Quantification was done using ImageJ (NIH) software. All the images are representative of the averaged results of the scoring (n = 3). Asterisks above a group define significance against the corresponding untreated control: * *p* < 0.05. Results are expressed as the mean ± standard deviation (SD).

**Figure 5 nutrients-17-00150-f005:**
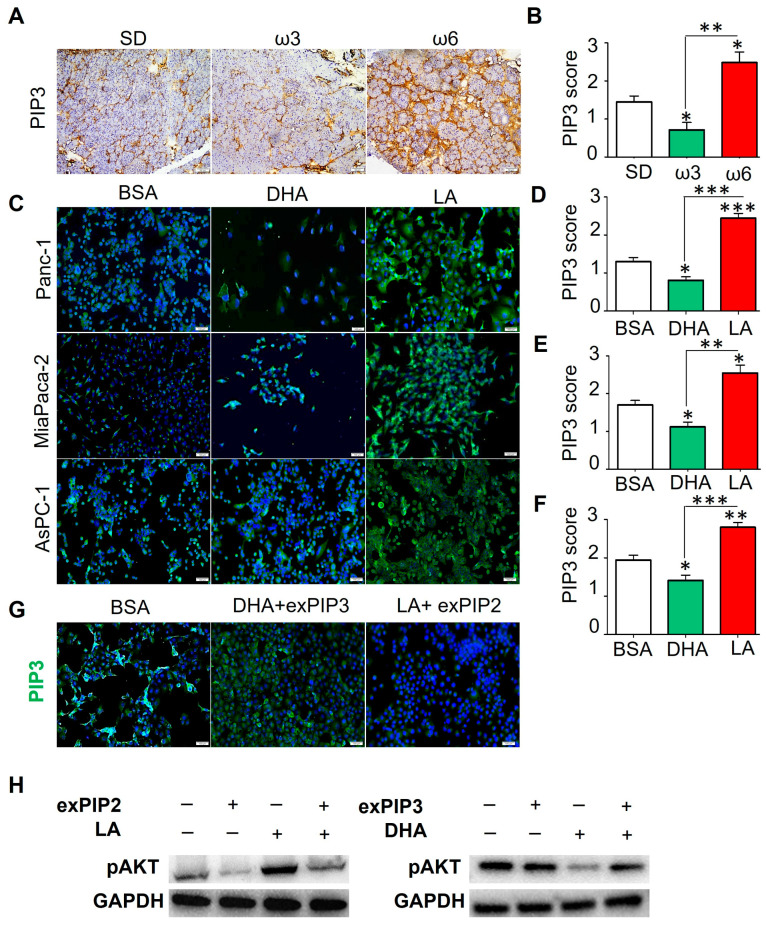
Dietary PUFAs modify the PIP2/PIP3 ratio in the membrane affecting AKT signaling. (**A**) Representative PIP3 IHC images of EK mice pancreata fed normal, ω3-, or ω6-enriched diets (n = 4). (**B**) PIP3 expression score from 0 to 3+, with 0 as no detectable immunostaining, 1 as 10–30% immunostaining, 2 as 30–60%, and 3 as >60%. The numerical score represents the average of 2 independent investigators. (**C**) Representative images of PIP3 immunocytochemical staining of PDAC cell lines incubated with DHA or LA at 40 µM for 48 h. PIP3 immunostaining score of (**D**) Panc-1, (**E**) MiaPaca-2, and (**F**) AsPC-1 cells. Staining was scored from 0 to 3+, with 0 as no detectable immunostaining, 1 as 10–30% immunostaining, 2 as 30–60%, and 3 as >60%. (**G**) Representative images of PIP3 immunocytochemical staining of the Panc-1 cell line incubated with BSA (left image), DHA 40 µM combined with 1 µM of exogenous PIP3 (middle image), and LA 40 µM combined with 1 µM of exogenous PIP2 (n = 3). (**H**) Western blot images of the Panc-1 pAKT levels after the exogenous supplementation of PIP2 and PIP3 to the PUFAs treatment. Results are expressed as the mean ± standard deviation (SD). Asterisks above a group define significance against the corresponding untreated control: * *p* < 0.05; ** *p* < 0.005; *** *p* < 0.0005. Results are expressed as mean ± standard deviation (SD).

**Figure 6 nutrients-17-00150-f006:**
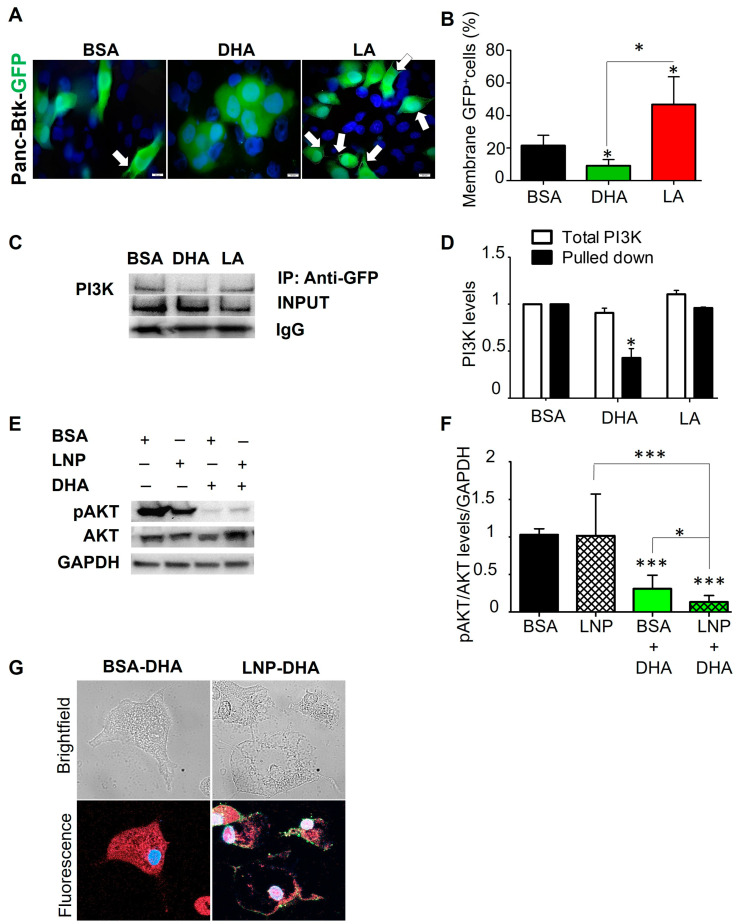
Dietary PUFAs modify PIP3 localization in the membrane, affecting AKT signaling, and exogenous PUFA administration by lipid nanoparticles (LNPs) improves PUFA delivery to the membranes. (**A**) Representative images of Panc-1 cells expressing the PH-BtK-EGF fusion protein (PIP3 biosensor) and incubated with 40 µM of DHA and LA. GFP (PIP3) expression was assessed with a confocal microscope. White arrows point to GFP-enriched spots at the plasma membrane. (**B**) Percentage of cells with membrane-positive staining relative to the total number of green cells (4 different fields of view, 20×). (**C**) Panc-1 cells transfected with GFP-C1-PLCdelta-PH were incubated with DHA and LA 40 µM for 48 h and subjected to immunoprecipitation. Western blot images of PI3k-alpha pulled down with GFP antibody to assess the binding of PI3K to PIP3. (**D**) Quantification of the replicates performed by immunoblotting. Quantification was done using ImageJ (NIH) software. All the images are representative of the averaged results of the scoring (n = 2). (**E**) The Panc-1 cell line was incubated with a lipid nanoparticle formulation (LNP) consisting of 90% GMO and 10% cholesterol with 40 µM DHA. The BSA group represents the control group treated with DHA bound to BSA (BSA.DHA) as a carrier. The LNP group represents the group treated with DHA encapsulated in LNP (LNP-DHA). Western blot images probing for total and phosphorylated AKT proteins. GAPDH was used as a loading control. (**F**) Quantification of the replicates performed by immunoblotting relative to the GAPDH expression level. (**G**) Representative images of confocal microscopy of Panc-1 cells treated with the complex BSA-DHA and with the complex LNP-DHA. Fatty acids are shown in red, LNP are shown in green, and the nucleus in blue. Asterisks in the graphs above a group define significance against the normal diet control: * *p* < 0.05; *** *p* < 0.0005. Bars indicate significant differences between two groups. Results are expressed as the mean ± standard deviation (SD).

## Data Availability

Data from this study is available from the corresponding author upon reasonable request.

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
