# Peer review of "Cell Membrane Fatty Acids and PIPs Modulate the Etiology of Pancreatic Cancer by Regulating AKT"

_nutrients, 2024, doi:10.3390/nu17010150_

Round 1
Reviewer 1 Report
Comments and Suggestions for Authors
The manuscript of Torres et al deals with the study of the effect of PUFA omega-3 (EPA and DHA), and their lipid nanoparticle preparation, in in vitro and in vivo models of pancreatic ductal adenocarcinoma (PDAC).
This group has already published in Nutrients 2018 in cells and animals of similar types the effects of omega-3 fatty acids (ref 51 cited in the Discussion), therefore in the Introduction (not in the Discussion) of this new manuscript the authors must evidence the unexplored parts of this research and what is the aim of the present study.
The extension of the importance of omega-3 obtained by this research to an evaluation of the consumption of omega-3 intakes using data available in population database (NHANES and NCI) and the frequency of PDAC seems to be not a very innovative approach. The inflammatory potential of diet and the risk of pancreatic cancer is very well known, and if there is a new aspect to consider this is not emerging from the present study.
The manuscript contains several critical points to be addressed before being considered for publication:
1. INTRODUCTION: the introduction must give an appropriate and updated overview of the role of this research. Why ref 52 is not cited here, since the approach is practically the same that is used in this investigation? In the introduction not only omega-6 and omega-3 must be cited, but also the role of saturated and MONOUNSATURATED (very important!) fatty acids that are present in the diet. Actually, what is important, also for the centrality of the membrane observation, is that the four types of fatty acids must have a balance in order to have impact on the Akt signaling, and the transfer of Akt to/from membrane spatiotemporally regulates its enzymatic functions. The activation of Akt upon binding PIP3 is a prerequisite for regulating cell proliferation, differentiation and migration (see Biophysical Reviews (2021) 13:123–138). What is determining the upregulation of Akt as seen in various tumors? Probably the fatty acid balance (not only the omega-3 and omga-6 levels) in cell membranes, to give an appropriate biophysical contribution, can give a more complete answer in the cell model chosen by the authors. The use of lipid-based nanoparticles (LNP) is reported in the introduction with the ref 20, which is a review listing a wide variety of “lipid containers”. In the introduction the appropriate references to the LNP used by the authors must be presented, in order to see if this is a novelty of their approach.
2. Materials and Methods and experimental choices: The access to a human database must be approved ethically, as it is needed for every observational study. What is the number of the approval?
A first remark is on experimental choices to use the membrane fatty acids analysis in cell models on one hand and the observation of the PIP3 staining in the pancreatic tissue of animal models on the other hand. If an important and new parallel can be done, why do not compare directly the membrane phospholipids of cell models with the membrane phospholipids of the pancreatic tissues? The extrapolation of cell membrane data with the PIP3 fluorescent levels seems to be more “indirect” than a direct comparison of the membrane compositions of the two models. This is a strong limitation of this study, and it is not understandable in a group that can do membrane isolation and has the two model in their hands.
A second remark is on LNP preparation which are not detailed as they must be, indicating the encapsulation efficiency and the release properties. Without this information, the LNPs formation and effect are not clear. If this preparation was used in other work, it must be reported appropriately.
3. Results
In Results the various parts have all the number 1. The it is not easy to indicate the results which are reported not appropriately.
FIRST RESULT: The first data on the population database must take into account the recent survey done in Int. J. Cancer. 2024;1–19 on the “Associations of plasma omega-6 and omega-3 fatty acids with overall and 19 site-specific cancers: A population-based cohort study in UK Biobank”. If the plasma levels of omega-6 and omega-3 can explain the results of the UK database, what is the impact to have the PUFA levels in membranes? The choice of plasma or membrane fatty acids in human research is still debated and the authors had the possibility in their animal model to compare both data. Instead, they did not provide evidences of the importance of membrane remodeling in animals, and this was a real weakness of their study.
SECOND RESULT: the animal models were fed standard diet (SD), whereas the other groups were fed the HFD + the PUFA. Where is the control group fed only HFD? As previously noted, in these animals the best result, and the most appropriate, would have been to provide the tissue phospholipid analysis to compare with the cell models.
FOURTH RESULT: the use of DHA in cells and the effect on cell viability is indicative of a membrane remodeling that occurs with a possible ferroptotic effect, present for DHA and absent for LA (which contains much less double bonds and also less peroxidizable that DHA). First of all, the complete membrane fatty acid content must be provided to show the remodeling of all fatty acid types. This is necessary for both the explanation of the viability effect and for the Akt remodeling. In cancer cell types, it was widely shown that full fatty acid remodeling occurs and this is an effect on the biophysical properties of membranes, as explained in the introduction, and can be extrapolated in humans as demonstrated with the RBC membrane lipidome (Annu Rev Physiol. 2019 Feb 10;81:165-188; Chem. Res. Toxicol. 2020, 33, 10, 2565–2572; Diagnostics 2017, 7(1), 1).
FIFTH RESULT: the phrase “These data suggest that composition of the phospholipids in the membranes of PDAC cells can be modified by dietary PUFA and that PUFAs are used, at least in part, to create new membranes in highly proliferative cells.” (page 9, 9 lines from the page end) must be eliminated or explained, since it is impossible to think that the authors do not know the lipid metabolism from dietary PUFA to PUFA in the cell membranes. PUFA are essential or semi-essential elements needed to form eukaryotic cell membranes, therefore their transfer from diet to membranes is a very well-known process.
4. Discussion: in many parts of the Discussion there is a parallel between the cells and animal models, which cannot be done without performing a parallel fatty acid analysis in cell membranes and tissue phospholipids.
The phrase “Previous work described phospholipid synthesis by tumor cells can be modified by dietary PUFAs[67] and that ω3 PUFA incorporation into phospholipids affects plasma membrane biophysical properties and alter recruitment/activation of signaling proteins[68] and lipid rafts and their downstream signaling cascades[62].” (page 12, 8 lines from the page end) is cited here, whereas it should have been the knowledge basis in the Introduction and a reason of the use of PUFA in this work, obviously using the membrane lipid analysis in all the experiments.
It is also evident that the authors are not aware, or do not consider, the efforts of the scientific community toward the “… mechanisms to demonstrate a direct means by which diet functions as a preventive strategy or even as a co-adjuvant treatment” as themselves say in their discussion (see review Int. J. Mol. Sci. 2022, 23(11), 6030). This is a pity since groups remain isolated instead of collaborating for the same objectives.
In general, the new results must be evidenced whereas, although relevant, the human database does not fit with the evidences of the cell and animal studies, and can be substituted with the reference to the recent survey appeared in Int. J. Cancer. 2024;1–19.
Author Response
3. Point-by-point response to Comments and Suggestions for Authors |
General comment: The manuscript of Torres et al deals with the study of the effect of PUFA omega-3 (EPA and DHA), and their lipid nanoparticle preparation, in in vitro and in vivo models of pancreatic ductal adenocarcinoma (PDAC). This group has already published in Nutrients 2018 in cells and animals of similar types of the effects of omega-3 fatty acids (ref 51 cited in the Discussion), therefore in the Introduction (not in the Discussion) of this new manuscript the authors must evidence the unexplored parts of this research and what the aim of the present study is.
The extension of the importance of omega-3 obtained by this research to an evaluation of the consumption of omega-3 intakes using data available in population database (NHANES and NCI) and the frequency of PDAC seems to be not a very innovative approach. The inflammatory potential of diet and the risk of pancreatic cancer is very well known, and if there is a new aspect to consider this is not emerging from the present study.
|
Response: We thank the reviewer for this observation and for pointing out the importance of highlighting our group’s prior work on this topic. In addition to the Nutrients 2018 study (previously ref 51), our laboratory has published other significant studies relevant to this field, which were already cited in the Introduction to provide a comprehensive background on our expertise and research trajectory. However, we acknowledge that the Nutrients 2018 publication is particularly relevant to the context of this manuscript. We have updated the Introduction to specifically reference this study, ensuring that it is properly integrated and that the unexplored aspects of the present research are clearly articulated. Additionally, as suggested latter on, we have included a brief explanation of the mechanism underlying AKT activation, emphasizing how modifications to membrane lipids—shaped by dietary lipid intake—can alter the structure of biological membranes and consequently impact signaling pathways that are directly orchestrated at the membrane, such as the PI3K-Akt pathway. We also appreciate the recommendation of the article Biophysical Reviews (2021) 13:123–138. This reference has enriched our work by providing a deeper biophysical perspective on membrane dynamics and their influence on signaling pathways, further supporting the context of our findings. Thank you for this valuable suggestion.
We have updated the introduction as follows (Pages 2-3):
Despite a low incidence of just over 1 in 10,000, pancreatic ductal adenocarcinoma (PDAC) remains one of the most aggressive solid malignancies with a 12% 5-year survival[1] compared to a 68% combined rate for all other cancers. PDAC has been projected to become the second cause of cancer related deaths in the next few years[2], which is mainly due to two factors: chemoresistance and late diagnosis[3]. Risk factors include family history, age, smoking, obesity, long-standing diabetes, chronic pancreatitis and diet[4], and recent evidence from genomic sequencing suggests a 15-year interval from initiation to aggressive disease. Both of these suggest a sufficient window of time for early detection[5] and cancer interception[6].
Though more research is needed for earlier detection, there is growing evidence that some cancers, PDAC among them, are influenced by diet and environmental factors[7]. Indeed, 30% of PDAC cases could be prevented by acting on modifiable risk factors (smoking, obesity, alcohol, and/or diet)[8]. Dietary fats, particularly their primary components—fatty acids—play a role in cancer development. However, the relationship between different fatty acid types, including saturated (SFAs), monounsaturated (MUFAs), and polyunsaturated (PUFAs), and PDAC risk remains a topic of ongoing debate and scientific discussion and the molecular mechanisms exploited by pancreatic cells to store and utilize fat to promote tumorigenesis are not well defined[9]. This has been studied in animal models to assess high fat diet (HFD) modification of mutant Kras-induced pancreatic neoplasia and cancer. This includes our seminal work comparing ω3[10] and ω6[11] PUFA-enriched diets with standard chow in EL-KrasG12D mice[12] and a similar approach with DHA and LA treatment also in human cells[13]. This study revealed a robust reduction in pAKT signaling and associated tumor suppression with ω3 diets, in contrast to the tumor-promoting effects of ω6 diets, highlighting their distinct contributions to pancreatic carcinogenesis. Additional research demonstrates that HFDs (predominantly animal fat/lard) activate Kras and Cox2 in acinar cells[14], reduce FGF21[15], and expand aerobic glycolysis to generate aggressive PDAC. Work with syngeneic orthotopic models generated similar findings[16] where a HFD (lard) increased cancer growth and fibrosis via CCKR signaling[17]. Yet, few have focused on HFDs enriched in PUFAs employing p48-Cre/LSL-KrasG12D(KC) mice[18] which well recapitulate Pancreatic Intraepithelial Neoplasia (PanIN) disease. Previous findings demonstrate that KC mice fed a HFD (corn oil based; 40% Kcal) promote more advanced pancreatic neoplasia[19] with increased incidence of PDAC, inflammation and dysregulated autophagy[20], peripancreatic visceral adiposity[21], and severe NK cell deficiencies.
To better understand the influence and mechanism of HFDs enriched in PUFAs on PDAC prevention, we investigated PUFA effects on disease progression employing both in vitro (PDAC cell lines) and in vivo (EL-Kras and KC mice) approaches. We evaluated the effects of HFDs supplemented with ω3 or ω6 PUFAs on lesion development. Our findings support that ω3-enriched HFDs lower lesion penetrance and cell proliferation associated with reduced pAKT, whereas a diet enriched in ω6-PUFA accelerated tumor formation and proliferation. Using PDAC cell lines incubated with docosahexaenoic acid (DHA), we observed reduced PI3K/AKT activity. This effect was reversed by Linoleic Acid (LA) and ω6-enriched HFDs, which can enhance pAKT levels. Akt activation is tightly regulated by its interaction with PIP3 at the membrane. PIP3, generated by phosphorylation of PIP2 by PI3K, recruits Akt to the membrane, where it undergoes phosphorylation and becomes active. This activation is crucial for promoting cellular processes such as proliferation, migration, and survival. Conversely, the enzyme PTEN dephosphorylates PIP3 to PIP2, releasing Akt from the membrane and deactivating this pathway. An imbalance in membrane lipid composition, potentially influenced by dietary fatty acids, can disrupt this regulation[22].
In physiological systems, fatty acids are predominantly transported through the bloodstream bound to serum albumin, which acts as their primary carrier, facilitating their solubility and delivery to tissues for metabolism, storage or incorporation into biological membranes[23]. To enhance bioavailability, solubility, and reduce degradation, FAs can be incorporated into lipid-based nanoparticles (LNPs)[24]. We have engineered FA-enriched LNPs, and similar results regarding inhibition of pAKT, a major downstream effector of Kras, were observed. This is encouraging because activating KRAS mutations are found in nearly 90% of PDAC cases[25], and it is considered an undruggable target though the KRASG12C inhibitor Sotorasib has FDA approval (ClinicalTrials.gov numbers, NCT04303780, NCT04185883 and NCT03600883) for adult subjects with advanced KRASG12C mutant expressing solid tumors. Yet, most prevalent in PDAC is KrasG12D which still has no effective inhibitors. Thus, targeting PI3K/AKT serves as a logical alternative.
Although discovery of early biomarkers and detection approaches are needed against PDAC, preventive strategies are equally required to reduce PDAC incidence and mortality. Building on our and others previous findings, in this work, we delve deeper into the molecular mechanisms underlying the observed effects of omega-3 fatty acids (DHA). Specifically, we propose a molecular mechanism responsible for the decreased proliferative capacity of pancreatic cells in the presence of DHA. Moreover, we developed lipid nanoparticles (LNPs) as a novel method to enhance PUFA delivery to the cell membrane, essentially altering lipid species content in the cell membrane. These LNPs demonstrated superior efficacy in reducing pAKT levels with ω3 PUFA compared to other standard carriers, such as BSA, underscoring the potential of leveraging biomembrane-based nanostructures to modulate cell signaling. The high biocompatibility and structural similarity of LNPs to cell membranes suggest promising applications for clinical translation in improving drug delivery and therapeutic efficacy. This study aims to not only expand our understanding of the distinct roles of dietary PUFAs in pancreatic cancer progression but also to explore innovative strategies for modulating these pathways, such as the use of lipid nanoparticles for potential clinical translation.
We also acknowledge that the association between the inflammatory potential of diet and the risk of pancreatic cancer is well-documented, and we recognize that the presented data are not novel in this regard. However, our intent was not to place these data at the center of our study, but rather to use them as a foundation for justifying the need to explore the effects of omega-3, and omega-6 fatty acids on pancreatic cancer incidence. These findings provide critical (human) context for our investigation. By establishing this foundational context, we aim to provide a strong basis for investigating underexplored aspects of dietary fats and their potential influence on pancreatic cancer risk.
The manuscript contains several critical points to be addressed before being considered for publication Response: We sincerely appreciate the detailed feedback provided. Following the point-by-point comments, we have carefully addressed each suggestion and incorporated the recommended changes into the manuscript where appropriate |
Comments 1: INTRODUCTION: The introduction must give an appropriate and updated overview of the role of this research. Why is ref 52 not cited here, since the approach is practically the same as that is used in this investigation?
Response: As stated above, we have modified the introduction to refer sooner to this work. It is now reference nº 13.
In the introduction not only omega-6 and omega-3 must be cited, but also the role of saturated and MONOUNSATURATED (very important!) fatty acids that are present in the diet. Actually, what is important, also for the centrality of the membrane observation, is that the four types of fatty acids must have a balance in order to have impact on the Akt signaling, and the transfer of Akt to/from membrane spatiotemporally regulates its enzymatic functions. The activation of Akt upon binding PIP3 is a prerequisite for regulating cell proliferation, differentiation and migration (see Biophysical Reviews (2021) 13:123–138). What is determining the upregulation of Akt as seen in various tumors? Probably the fatty acid balance (not only the omega-3 and omga-6 levels) in cell membranes, to give an appropriate biophysical contribution, can give a more complete answer in the cell model chosen by the authors.
Response: We thank the reviewer for their insightful comments regarding the diversity of fatty acids in the diet. While it is true that dietary fats include a variety of fatty acids, such as saturated, monounsaturated, and polyunsaturated fatty acids (PUFAs) (and we have included a reference regarding this, ref nº 9), our work specifically focuses on the role of PUFAs in pancreatic tumor cell proliferation. The animal diets used in our study were enriched predominantly in PUFAs—either omega-3 or omega-6 fatty acids (Supplementary Table 1 and 2). Additionally, all in vitro experiments were conducted using DHA (an omega-3 fatty acid) and LA (an omega-6 fatty acid) (Supplementary Figure 3). This targeted approach allows us to investigate the specific molecular effects of these PUFAs on tumor cell signaling and proliferation, particularly through their influence on membrane composition and the PI3K-Akt signaling pathway. Therefore, our study emphasizes the unique role of PUFAs rather than exploring the broader context of all dietary fatty acids.
The use of lipid-based nanoparticles (LNP) is reported in the introduction with the ref 20, which is a review listing a wide variety of “lipid containers”. In the introduction the appropriate references to the LNP used by the authors must be presented, in order to see if this is a novelty of their approach.
Response: We thank the reviewer for this comment. We updated the references and clarified that portion of the text to include more details about LNP formulation rationale. The text now reads:
To enhance bioavailability, solubility, and reduce degradation, FAs can be incorporated into lipid-based nanoparticles [28]. We used a formulation based on cholesterol and glycerol monooleate as it has been demonstrated to be stable and have high fusogenic affinity to target membranes [L. Zheng, S. Bandara, and C. Leal*, Proc. Natl. Acad. Sci. USA. 120, 27, e2301067120 (2023)]
|
Comments 2: Materials and Methods and experimental choices: The access to a human database must be approved ethically, as it is needed for every observational study. What is the number of the approval?
Response: According to federal regulations governing research involving human subjects (45 CFR Part 46), Institutional Review Board (IRB) approval is not required for research utilizing publicly available datasets, provided that:
Given that our study adheres to these criteria, IRB approval was not applicable or sought for this research.
We have added this information in material and methods (page 4):
Observational study: According to federal regulations governing research involving human subjects (45 CFR Part 46), Institutional Review Board (IRB) approval is not required for research utilizing publicly available datasets, provided that the datasets are publicly accessible, and the data are de-identified, uncoded, and stripped of all identifiers. Given that our study adheres to these criteria, IRB approval was not applicable or sought for this research.
A first remark is on experimental choices to use the membrane fatty acids analysis in cell models on one hand and the observation of the PIP3 staining in the pancreatic tissue of animal models on the other hand. If an important and new parallel can be done, why do not compare directly the membrane phospholipids of cell models with the membrane phospholipids of the pancreatic tissues? The extrapolation of cell membrane data with the PIP3 fluorescent levels seems to be more “indirect” than a direct comparison of the membrane compositions of the two models. This is a strong limitation of this study, and it is not understandable in a group that can do membrane isolation and has the two model in their hands.
Response: We appreciate the reviewer’s insightful comment regarding the potential to directly compare membrane phospholipid compositions between cell models and pancreatic tissues. While we agree that such an analysis would provide valuable information, this approach would be far beyond the scope of our initial objectives and the resources available for this study. Our original focus was on understanding regulatory mechanisms at the protein level, and the potential role of membrane lipids only emerged as the study progressed. As a result, when our findings pointed toward a membrane-related mechanism, we aimed to explore this as thoroughly as possible with the samples and funding available. Since the primary membrane phospholipid we have explored in this work is phosphatidylinositols (PIs), garnering enough tissue/cell samples is not trivial given that PIs are dramatically lower than other phospholipids (phosphatidylcholine, phosphatidylethanolamine, phosphatidylserine, and others). Yet to the reviewer’s point, we acknowledge this as a limitation of our study and are now considering this an important direction for future research. We agree that performing detailed phospholipid and other lipid analyses on diet -treated pancreatic tissues and cells would expand this field of research and offer a plethora of lipid interactions/associations worth investigation. Indeed, a direct comparison of membrane phospholipid compositions in both models would undoubtedly strengthen the link between dietary PUFAs, membrane composition, and Akt signaling. We appreciate the reviewer’s perspective and hope this explanation provides clarity regarding our methodological choices.
A second remark is on LNP preparation which are not detailed as they must be, indicating the encapsulation efficiency and the release properties. Without this information, the LNPs formation and effect are not clear. If this preparation was used in other work, it must be reported appropriately.
Response: We added more information about LNP preparation methods in the LNP preparation method. The text now reads:
LNPs were prepared by standard methods based on nanoprecipitation known to yield uniform distribution of size, shape, and encapsulation efficiency [L. Zheng, S. Bandara, and C. Leal*, Proc. Natl. Acad. Sci. USA. 120, 27, e2301067120 (2023), Musielak, E.; Feliczak-Guzik, A.; Nowak, I. Synthesis and Potential Applications of Lipid Nanoparticles in Medicine. Materials (Basel) 2022, 15 (2), 682, DOI:10.3390/ma15020682]. Briefly, LNPs were prepared in a NanoAssemblr Ignite (Precision NanoSystems) by microfluidics. The desired ratio of lipids (with/without PUFA) was dissolved in chloroform. After full removal of the organic solvent overnight under a N2 flow, the lipids were then dissolved in ethanol at a concentration of 10 mM. The total flow rate was maintained at 12 mL per min and a 4:1 ratio of aqueous to ethanol inlets. After removal of ethanol under dialysis for 24h the resulting formulation of LNPs dispersed in aqueous buffer was 90% glyceryl monooleate (Sigma-Aldrich) and 10% cholesterol (Sigma-Aldrich) with 40/80 μM DHA/LA. The particles were then added to culture media to stabilize at 37°C overnight prior to cell treatment. DiO (Invitrogen) for lipid tracing and FA dye was added along with the DHA treatment and used for confocal imaging (LSM800, Zeiss).
Comments 3: Results. In Results the various parts have all the number 1. The it is not easy to indicate the results which are reported not appropriately.
Response: We apologize for the formatting error in the numbering of the results section, which caused all the results to appear as number 1. This issue has now been corrected in the revised version of the manuscript. Thank you for bringing this to our attention.
FIRST RESULT: The first data on the population database must take into account the recent survey done in Int. J. Cancer. 2024;1–19 on the “Associations of plasma omega-6 and omega-3 fatty acids with overall and 19 site-specific cancers: A population-based cohort study in UK Biobank”. If the plasma levels of omega-6 and omega-3 can explain the results of the UK database, what is the impact to have the PUFA levels in membranes? The choice of plasma or membrane fatty acids in human research is still debated and the authors had the possibility in their animal model to compare both data. Instead, they did not provide evidences of the importance of membrane remodeling in animals, and this was a real weakness of their study.
Response: Thank you for your comment. We have incorporated reference to Associations of plasma omega-6 and omega-3 fatty acids with overall and 19 site-specific cancers: A population-based cohort study in UK Biobank, as it aligns well with the findings from the NHANES database and strengthens the evidence supporting the association between a higher omega-6/omega-3 ratio and increased risk. It is now reference nº55. We sincerely thank the reviewer for bringing this important study to our attention.
While we understand the value of measuring fatty acids in blood as biomarkers of dietary intake, this was not the aim of our study. Our objective was not to evaluate the type or amount of fats being consumed by the animals but rather to investigate the phenotypic effects of well-characterized diets, where the predominant fatty acids (PUFAs, either ω3 or ω6) were already well-defined. Knowing the detailed composition and overall impact on disease progression of these diets allowed us to focus on assessing the contribution of these dietary fatty acids on mechanisms related to Akt signaling in mice, rather than tracking their presence in blood. Though we appreciate this avenue of research and how it would marry well with the human data, the amount of blood required and the storage thereof (need to prevent lipid perioxidation in much smaller aliquots of mouse blood) would need to be empirically established using large cohorts of mice fed these PUFA-enriched diets. We hope this explanation clarifies the rationale behind our approach.
SECOND RESULT: the animal models were fed standard diet (SD), whereas the other groups were fed the HFD + the PUFA. Where is the control group fed only HFD? As previously noted, in these animals the best result, and the most appropriate, would have been to provide the tissue phospholipid analysis to compare with the cell models.
Response: Thank you for the comment. To clarify, the standard diet (SD) used in our study is equivalent in macronutrient composition and caloric density to the high-fat diets (HFDs), with the sole difference being the type and proportion of polyunsaturated fatty acids (PUFAs) included. Specifically:
● The standard diet includes 177.5 g of lard and 25 g of soybean oil. ● The ω6-enriched diet replaces lard with 189.5 g of safflower oil (a rich source of linoleic acid, an omega-6 PUFA). ● The ω3-enriched diet replaces lard with 189.5 g of menhaden oil, which is high in docosahexaenoic acid (DHA) and eicosapentaenoic acid (EPA), both omega-3 PUFAs(Supplementary Table 2). This design ensures that the primary difference between the diets is the predominant type of PUFA, allowing us to specifically evaluate the effects of omega-3 and omega-6 fatty acids on disease progression.
We hope this detailed explanation, supported by the data provided in the supplementary materials, addresses your concern. Thank you for giving us the opportunity to clarify this aspect of our methodology.
FOURTH RESULT: the use of DHA in cells and the effect on cell viability is indicative of a membrane remodeling that occurs with a possible ferroptotic effect, present for DHA and absent for LA (which contains much less double bonds and also less peroxidizable that DHA). First of all, the complete membrane fatty acid content must be provided to show the remodeling of all fatty acid types. This is necessary for both the explanation of the viability effect and for the Akt remodeling. In cancer cell types, it was widely shown that full fatty acid remodeling occurs and this is an effect on the biophysical properties of membranes, as explained in the introduction, and can be extrapolated in humans as demonstrated with the RBC membrane lipidome (Annu Rev Physiol. 2019 Feb 10;81:165-188; Chem. Res. Toxicol. 2020, 33, 10, 2565–2572; Diagnostics 2017, 7(1), 1).
Response: Thank you for the comment. While we agree that analyzing the complete fatty acid content of membranes would enhance our understanding of the remodeling process, this approach, as mentioned in regards to the complete phospholipid profile of these cell membranes, would require a large animal cohort administered these diets followed by lipidomics. This would be a separate study perhaps in combination with the phospholipid profiling that would generate multiple points of inquiry beyond the scope of this current study.
Specifically, while we assessed DHA incorporation into phospholipids such as phosphatidylcholine (PC), phosphatidylethanolamine (PE), and triglycerides (TG) in Panc-1 cell membranes using GC-MS after TLC separation, we were unable to measure phosphatidylinositol (PI) due to its low abundance in membranes, which exceeds the detection sensitivity of our method. These limitations also influenced our ability to provide a comprehensive analysis of all fatty acid types in membranes. However, our data demonstrates that DHA treatment significantly increases its incorporation into PC, PE, and TG in a dose-dependent manner, supporting the idea that PUFAs from the diet contribute to membrane building and remodeling in proliferative cells. These findings are discussed in the manuscript and shown in Supplementary Figure 2 C-E. Additionally, we visualized membrane PIP3 levels through immunofluorescence in pancreatic tissues from omega-3 HFD-fed mice. These analyses revealed significant reductions in membrane PIP3 staining compared to omega-6 HFD and SD-fed mice, highlighting functional differences associated with dietary PUFAs. Finally, we conducted an affinity assay via immunoprecipitation to further explore PIP3-related signaling. However, we acknowledge that these are indirect methods and that a more comprehensive membrane fatty acid profiling approach, as you suggested, would provide further supporting evidence along with other lipid alterations in cell membrane worth consideration.
We recognize the need for membrane lipidomics in future experiments and are incorporating plans in forthcoming grant applications to conduct full membrane lipidomics from the outset, ensuring tissue samples are stored and processed under conditions that allow the application of these more advanced techniques. We appreciate the reviewer’s thoughtful feedback and agree that this direction would significantly strengthen the study design and outcomes in future research. Thank you for highlighting this important point.
FIFTH RESULT: the phrase “These data suggest that composition of the phospholipids in the membranes of PDAC cells can be modified by dietary PUFA and that PUFAs are used, at least in part, to create new membranes in highly proliferative cells.” (page 9, 9 lines from the page end) must be eliminated or explained, since it is impossible to think that the authors do not know the lipid metabolism from dietary PUFA to PUFA in the cell membranes. PUFA are essential or semi-essential elements needed to form eukaryotic cell membranes, therefore their transfer from diet to membranes is a very well-known process.
Response: Thank you for your thoughtful feedback. We fully agree that the incorporation of PUFAs into cell membranes is a well-established process and not the primary focus of our study. What we intended to highlight is that in our models, the fundamental effect of dietary PUFAs appears to be driven by their incorporation into cell membranes and its subsequent impact on signaling pathways, particularly the PI3K-Akt pathway. While PUFAs trigger a wide array of pleiotropic effects on numerous cellular processes, including inflammation, gene expression, and cell survival, our study specifically investigates how their incorporation in the membrane influences Akt signaling in pancreatic cancer cells. We did not aim to uncover the well-known fact that fatty acids are incorporated into membranes, but rather to focus on the unique impact that this incorporation has on the membrane's biophysical properties and how these properties subsequently affect downstream signaling events in the context of cancer. We will revise the statement to better reflect this distinction and ensure that it aligns with the specific aims of our study. We appreciate your feedback and will have adjusted the text accordingly (page 9):
Based on these data, it appears that among the various potential effects that dietary PUFAs can have on cellular physiology, in our model, the primary mechanism involves their incorporation into cell membranes. This incorporation seems to play a pivotal role in triggering the observed changes in signaling pathways, particularly in the PI3K-Akt cascade, rather than other pleiotropic effects commonly associated with fatty acids. These findings highlight the significance of membrane remodeling as a central driver of the phenotypic effects seen in pancreatic cancer cells.
Comments 4: Discussion: in many parts of the Discussion there is a parallel between the cells and animal models, which cannot be done without performing a parallel fatty acid analysis in cell membranes and tissue phospholipids.
Response: We appreciate the reviewer’s observation regarding the parallel drawn between cell and animal models. We acknowledge that a direct comparison would ideally require fatty acid analysis of cell membranes and tissue phospholipids to validate the parallels. While such an analysis was not possible in the present study, we based our discussion on previously reported data that link cellular and tissue responses in comparable contexts. To ensure clarity, we have taken care throughout the manuscript to specify whether the results discussed are derived from in vitro studies or animal models. Nonetheless, we have revised the discussion further to highlight this distinction explicitly and to better reflect the exploratory nature of the parallels we propose, while acknowledging the limitations stemming from the lack of direct comparative analyses.
The phrase “Previous work described phospholipid synthesis by tumor cells can be modified by dietary PUFAs[67] and that ω3 PUFA incorporation into phospholipids affects plasma membrane biophysical properties and alter recruitment/activation of signaling proteins[68] and lipid rafts and their downstream signaling cascades[62].” (page 12, 8 lines from the page end) is cited here, whereas it should have been the knowledge basis in the Introduction and a reason of the use of PUFA in this work, obviously using the membrane lipid analysis in all the experiments.
Response: We agree with the reviewer that these references are more appropriate in the introduction. Therefore, we have moved the corresponding paragraph from the discussion to the introduction to better align with the manuscript's structure and context. Thank you for this helpful suggestion.
It is also evident that the authors are not aware, or do not consider, the efforts of the scientific community toward the “… mechanisms to demonstrate a direct means by which diet functions as a preventive strategy or even as a co-adjuvant treatment” as themselves say in their discussion (see review Int. J. Mol. Sci. 2022, 23(11), 6030). This is a pity since groups remain isolated instead of collaborating for the same objectives. In general, the new results must be evidenced whereas, although relevant, the human database does not fit with the evidences of the cell and animal studies, and can be substituted with the reference to the recent survey appeared in Int. J. Cancer. 2024;1–19.
Response: Thank you for bringing this broader perspective to our attention. We fully agree that dietary fatty acids hold significant potential for supporting cancer therapies through their pleiotropic effects on cellular physiology and their role as modulators of membrane composition and signaling pathways. The review (Int. J. Mol. Sci. 2022, 23(11), 6030) provides a comprehensive framework that complements our findings, and we appreciate the opportunity to integrate its insights into our discussion.
Fatty acids are increasingly recognized for their importance in biological, pharmacological, and clinical contexts. Their roles span from structural and functional contributions at the cellular level to nutritional and therapeutic applications, including as biomarkers of cancer onset and progression. As highlighted in the review, their essentiality, oxidizability, and involvement in lipogenesis and desaturase pathways make them critical targets for both fundamental research and translational approaches in cancer therapy. The review also emphasizes the central role of membrane fatty acid composition for membrane lipid therapy and the synergistic potential of combining fatty acid-based foods and nutraceuticals with pharmacological interventions. This multidisciplinary approach underscores the necessity of a holistic strategy, integrating nutra- and pharma-strategies to advance cancer research and therapy.
Our study aligns with these principles by focusing on how dietary PUFAs, specifically omega-3 and omega-6 fatty acids, influence membrane phospholipids (primarily PIs) of pancreatic cancer cells, which in turn modulates critical signaling pathways such as PI3K-Akt. While the incorporation of fatty acids into membranes is a well-known process, our work demonstrates that, in our model, this incorporation drives key phenotypic effects, particularly in the regulation of Akt signaling and cell viability. These findings highlight the specific impact of membrane remodeling in pancreatic cancer, supporting the idea of membrane-targeted approaches in cancer treatment.
We acknowledge the importance of personalizing fatty acid-based strategies, as the review suggests, and recognize that our study represents one piece of a larger puzzle in understanding how dietary components can support pharmacological therapies. To better align with the broader scientific context, we have revised our discussion to reflect these multidisciplinary and translational aspects, incorporating the concepts outlined in the review to strengthen the positioning of our work within this growing field.
We sincerely thank the reviewer for directing us to this valuable resource and for encouraging us to better integrate our findings into the broader scientific efforts in cancer therapy and prevention.
The new discussion reads as follows (page 12):
PDAC incidence and mortality continues to rise[8,46] despite advancement in the knowledge of its etiology and risk factors. Early detection is paramount and would have a profound benefit for patient survival[47] but would require effective preventative measures to interrupt tumor progression[3], which is possible with more than a decade of time be-tween neoplasia and advanced disease[48]. A number of such environmental factors include tobacco exposure, alcohol use, diabetes, chronic pancreatitis, obesity, and diet. With an epidemiological association between PDAC incidence and mortality in developed countries, having a prevalence for HFDs is likely a starting point for dietary interventions[8]. Indeed there is decreased risk with consumption of fruit, vegetables, and folate-rich foods[49,50] and increased risk associated with the intake of red and processed meat[51]. These findings have been well-tested in several PDAC mouse models. There has been a prominent turn from animal fat to a higher intake of ω6 PUFAs due to their cardio-vascular benefit. Yet, it is important to assess the type of PUFA on other diseases including cancer. The Healthy Lifestyle Index (HLI) combines information on smoking, alcohol intake, dietary habits, BMI, and physical activity and has been related to PDAC within the EPIC study[52], which shows that adherence to a combination of healthy lifestyle habits was strongly inversely associated with PDAC risk. Epidemiological studies are crucial to learn population behavior and disease patterns, particularly for relative low incidence pathologies such as pancreatic cancer. Several epidemiological studies associate increased PDAC incidence with dietary fat and obesity[53–56]. The biological basis underlying this association still requires further study, with a specific focus on the role of each individual dietary fat. Our findings align with the broader understanding of PDAC as a multifactorial disease influenced by lifestyle and dietary factors.
A hallmark of cancer cells, including PDAC, is their ability to sustain unlimited proliferation, which requires substantial metabolic and structural resources. Fatty acids, particularly PUFAs, serve not only as membrane components but also as substrates for energy metabolism and mediators of cell signaling [57]. Thus, lipogenesis is accelerated in cancer cells, as high proliferative rates lead to preferential fatty acids uptake from the environment[58]. PDAC cells have been shown to utilize exogenous fatty acids and stored lipids to enhance their metastatic capacity [59].
Socioeconomic changes likely contribute to the change in the proportion of fats ingested in the diet. Changing dietary habits have led to a change in fatty acid consumption, with an increase in ω6 fatty acids and a marked reduction in the consumption of ω3 fatty acids. In addition, the ω6 FAs content increased considerably as a result of the rise (up to 70%) in ω6 rich grains and also the addition of vegetable oil. This in turn has resulted in an imbalance in the ω6/ω3 ratio, very different from the original 1:1 ratio that humans had in the past[60]. In this context, this imbalance has been highlighted as a factor contributing to disease development and progression. Both the analysis of data from the NHANES population database and findings from a recent study consistently indicate that a higher ω6/ω3 ratio is associated with an increased risk of certain health conditions[61]. These results underline the importance of dietary balance between these fatty acids and provide further evidence for the role of ω3 and ω6 polyunsaturated fatty acids in disease development and prevention. Our study contributes to this field by demonstrating that ω3 and ω6 PUFAs play a central role in PDAC progression in our model, primarily through their modulation of Akt signaling pathways. While PUFAs are known to influence various cellular processes, our findings suggest that their impact on Akt signaling is the predominant mechanism affecting tumor cell proliferation and survival in this context [13,62,63]. AKT is a serine-threonine protein kinase with roles in cell growth, proliferation and apoptosis in a PI3K-dependant manner. PI3K, its substrate (PIP2), and product (PIP3) are considered obligate and rate-limiting for proper AKT activation. Inactive cytosolic AKT is recruited to the membrane and engages PIP3 through AKT PH domain binding leading to phosphorylation and activation of AKT (reviewed in[64]). In this work, we focused on how PUFAs modify these functional upstream modulators of AKT by altering configuration of mem-brane phospholipids and subsequent changes in pAKT leading to promotion or inhibition of cell proliferation. This preventive effect was confirmed in two mouse models with different degrees of lesion histotype (cystic papillary neoplasms or CPNs and PanINs) and kinetics. ω3 HFD was able to reduce progression and development of neoplastic lesions in these animals with a substantial reduction in lesion frequency, cell proliferation, and fibrosis. Under condition of excess ω6 FA-enriched HFDs, a prevalence of neoplastic lesions was identified: a substantial increase in frequency with a higher proliferative index and more fibrotic tissue. Although the difference between ω3 HFD and SD was significant, more dramatic change was obtained between the two HFD groups. Other studies based on ω3 HFD in animal models of carcinoma have described a reduction in tumor size and improved sur-vival[65,66]. Fat-1-p48Cre/+-LSL-KrasG12D/+ exhibited a dramatic inhibition of PDAC incidence, frequency of PanIN-3 formation, and their progression to PDAC[67]. Fat-1 mice convert ω6 to ω3 FAs which can mitigate potential confounding dietary factors. Orthotopic PDAC mice fed a HFD (enriched in oleic and linoleic acids) had increased tumor growth and metastasis[16]. Our results suggest that DHA-enriched diets mitigate neoplastic progression in mouse models, supporting the potential of ω3 PUFAs as co-adjuvants in cancer therapy.
Incubation of cancer cell lines with DHA demonstrated multiple anticancer mechanisms of action including inhibition of cell proliferation[68,69], migration, invasion[70] and apoptosis-inducing capacity[20,71,72]. Our in vitro results with PDAC cells confirm an antiproliferative effect of DHA concomitant with a reduction in the activation of AKT-mediated proliferation and subsequent reduction of downstream AKT effectors, BAD and FOXO3a. We observed an increase in total levels of BAD which interacts with either Bcl-2 or Bcl-XL neutralizing their anti-apoptotic functions[73]. DHA also induced total levels of unphosphorylated FOXO3a which is a tumor suppressor factor. Recently, it has been shown that FOXO3a expression was remarkably reduced in PDAC tissues, and correlated with metastasis-associated pathologic characteristics and poor prognosis in PDAC patients[74]. Hence, the inhibition of PDAC cell proliferation can be mediated by DHA-induction of FOXO3a as others have described previously[75]. Interestingly, the protective role of ω3 seems to be more prevalent when both fats are present.
With no significant difference in levels of regulators of AKT phosphorylation, we aimed to study if PUFA cell membrane incorporation altered AKT-regulated signals. We observed that supplemental DHA influenced the plasma membrane composition, aligning with previous studies that have demonstrated its potential role in modifying phospholipid profiles. These findings support the idea that DHA can affect cellular processes by altering membrane-associated signaling mechanisms, consistent with established research[76–78]. DHA treatment increased while LA reduced the PIP2:PIP3 ratio in PDAC cell membranes. DHA likely modifies the structure of PIP2, reducing its affinity for PI3K which is required to convert PIP2 into PIP3[79]. The dissociation of PI3K from PIP2 prevents further production of PIP3 which prevents AKT phosphorylation and activation, thereby disabling AKT signaling. The proposed mechanism is depicted in supplementary figure 3C. Since PUFAs appear to be pleiotropic, impacting metabolic and non-metabolic pathways, it is essential to further dissect all these mechanisms to demonstrate a direct means by which diet functions as a preventive strategy or even as a coadjuvant treatment. ω3 PUFAs have been shown to act synergistically with some chemotherapeutic or chemopreventive agents[69,80]. Our results support this idea since the antiproliferative capacity of gemcitabine in vitro was enhanced when administered in combination with DHA, even at lower concentrations. This reduced the dose of gemcitabine necessary to inhibit growth in cell culture by 50%, suggesting this approach could be amenable to an additional anti-cancer agent. In a recent study investigating intravenous ω3 PUFAs and gemcitabine chemotherapy versus gemcitabine therapy only in patients with PDAC, the combined treatment significantly reduced immune modulatory cells (particularly myeloid derived suppressor cells) and stability of Tregs[81].
One area that needs exploration would be related to optimizing the administration of PUFAs to minimize side effects that may occur such as abdominal pain or diarrhea and allowing the rapid incorporation into cell membranes. With the idea of improving the delivery method of ω3 PUFA to pancreatic cells, we have developed DHA-containing lipid nanoparticles (LNPs). Our findings indicate that DHA-LNPs can be utilized for PUFA uptake in cells and impact AKT-regulated proliferation. Modification of the lipid profile of liposomes has been shown to directly have an impact on the proliferation of breast cancer cells incubated with them[82]. A low-density lipoprotein (LDL)-based nanoparticles with DHA was engineered to enhance physical, oxidative stability, and delivery of DHA to target cells achieving selective cytotoxicity toward hepatocarcinoma cells and enhanced tumor cell death through ferroptosis[83,84]. Application of ω3 PUFA nanoformulations in lung and prostate cancer demonstrated that the simultaneous presence of DHA in the nanoparticles enhanced the anticancer activity of taxanes both in vitro and in vivo, and also increased the mean survival time of mouse models[28]. More research needs to be done, especially in PDAC including in vivo evaluations of nanoformulations in the preclinical setting, especially in combination with one or more chemotherapeutic agent within these LNPs. Besides, phospholipid and fatty acid lipidomics represent valuable future avenues of research to confirm and extend our findings. These approaches could provide deeper insights into the specific lipid species involved in modulating signaling pathways and membrane dynamics, further validating the observed effects of dietary PUFAs on pancreatic cancer progression. |

Reviewer 2 Report
Comments and Suggestions for Authors
Pancreatic ductal adenocarcinoma (PDAC) is one of the worst tumor types due to its relatively low 5-year survival rate, late diagnostics, and therapeutic difficulties. In the manuscript by Torres et al., the effects of polyunsaturated fatty acids (PUFA) on disease progression were analyzed. The research was conducted based on database analyses and in vitro and in vivo experiments. The general aim was to understand how a diet rich in ω3 and ω6 polyunsaturated fatty acids affects the progression of PDAC.
Consumption of PUFAs in a patient population correlates with increased PDAC incidence, where mostly ω6 intake increases the risk. The PUFA exerts an effect on PDAC cells via membrane structural modifications, most likely from the incorporation of these fatty acids into phospholipids including PIP2 and potentially altering PIP2-PI3K interactions. Proper dietary interventions can prevent the development of PDAC but also can improve the antiproliferative effect of drug efficacy during treatment.
The study is very interesting and worth attention. Both strengths and weaknesses were discussed.
In my opinion, only editorial corrections are needed:
1. Adapting the layout and style of the manuscript to the journal's requirements.
2. Correction of Abstract - the word limit is max 200; please do not use Headings in Abstract, like Background, Results, ...
3. Please, add the number and date of Ethical Committee Aprovment for in vivo experiments.
Author Response
3. Point-by-point response to Comments and Suggestions for Authors |
Comments 1: Pancreatic ductal adenocarcinoma (PDAC) is one of the worst tumor types due to its relatively low 5-year survival rate, late diagnostics, and therapeutic difficulties. In the manuscript by Torres et al., the effects of polyunsaturated fatty acids (PUFA) on disease progression were analyzed. The research was conducted based on database analyses and in vitro and in vivo experiments. The general aim was to understand how a diet rich in ω3 and ω6 polyunsaturated fatty acids affects the progression of PDAC. Consumption of PUFAs in a patient population correlates with increased PDAC incidence, where mostly ω6 intake increases the risk. The PUFA exerts an effect on PDAC cells via membrane structural modifications, most likely from the incorporation of these fatty acids into phospholipids including PIP2 and potentially altering PIP2-PI3K interactions. Proper dietary interventions can prevent the development of PDAC but also can improve the antiproliferative effect of drug efficacy during treatment. The study is very interesting and worth attention. Both strengths and weaknesses were discussed.
In my opinion, only editorial corrections are needed: 1. Adapting the layout and style of the manuscript to the journal's requirements. 2. Correction of Abstract - the word limit is max 200; please do not use Headings in Abstract, like Background, Results, ... 3. Please, add the number and date of Ethical Committee Approvement for in vivo experiments.
|
Response 1: We sincerely thank the reviewer for their thoughtful and constructive feedback. We are pleased to know that the study was found to be of interest and value. In response to the editorial comments:
Manuscript layout and style: We have carefully reviewed the manuscript to ensure it adheres to the journal’s formatting and style guidelines.
Abstract: The abstract has been revised to comply with the journal's word limit and format. We have followed the guidance provided on the journal’s official webpage, which specifies that research articles should include a structured abstract of up to 250 words with the following headings: Background/Objectives, Methods, Results, and Conclusions.
Ethical Committee Approval: We have added the approval number and date of the Ethical Committee for the in vivo experiments to the manuscript, as requested (Page 4):
Observational study: According to federal regulations governing research involving human subjects (45 CFR Part 46), Institutional Review Board (IRB) approval is not required for research utilizing publicly available datasets, provided that the datasets are publicly accessible, and the data are de-identified, uncoded, and stripped of all identifiers. Given that our study adheres to these criteria, IRB approval was not applicable or sought for this research. Animals and diets: All experiments involving the use of mice were performed following protocols approved by the Institutional Animal Care and Use Committee at the University of Illinois at Chica-go (UIC), Approval Code: 14-138, Approval Date: August 28,2014.
Thank you again for your helpful suggestions, which have improved the clarity and compliance of the manuscript.
|

Reviewer 3 Report
Comments and Suggestions for Authors
The manuscript entitled “Cell membrane fatty acids and PIPs modulate the etiology of pancreatic cancer by regulating AKT”, by Carolina Torres et al. as research article, was submitted to Nutrients for possible consideration to be published.
PDAC is one worst solid malignancy regarding the outcomes and metabolic dysfunction leading to cachexia. In this study, the authors reported the importance of dietary intervention in treating this disease. The dietary PUFAs like ω3-FA can incorporate into plasma membrane lipids to affect PI3K/AKT signaling and support the emergence of membrane-targeted therapies, as so in this study, abrogate PI3K activity thereby reduce AKT activation and ultimately suppress cell proliferation. Phospholipids in lipid nanoparticles (LNP) impact cell membrane integrity and ultimately cell viability after administration.
Overall, the study was nice but quite many typos and some major issues should be carefully addressed.
Some issues:
1) Regarding the Cell viability assay, I read that each well cell message as authors wrote 5 x 104 cells/ml were used but how about the number of cells per well in the experiments?
2) “hours”, “h”, “hr”, “hour”, and “hrs” were used and mixed. Not uniform.
3) “10% formalin” means what kind concentrations? “1% BSA” means what then? And “5% β-mercaptoethanol”, “0.1% SDS”, “0.05% Tween 20”?
4) “40/80 μM DHA/LA” was not standard and clear.
5) “Values are expressed as ± SD.”
6) “The significant differences were indicated by *p<0.05; **p<0.005; ***p<0.0005”, and “The specific p<0.05 was considered statistically significant.”
7) Results section: how to number the results?
8) Figure 2G-H, omega-6 should be more significant that omega-3 for the p-ERK/ERK. But in the figure, both were very similar, and no big difference, compared to SD group.
9) Figure 3DEF, 2.5 uM LA showed a sudden increase of viable cells. The cell strain should be labeled in Figure G. The X-axis of Figure H was no correct.
10) Figure 4D, 40 uM LA treatment was underestimated in Fig. 4G.
Figure 4B, 5 uM DHA treatment was underestimated in Fig. 4H.
Figure 4F, 40 uM LA treatment was underestimated in Fig. 4I.
11) In terms of the quantitative analysis, I am not sure the results in Figure 6C and D? Figure 6 E and F? BSA group = LNP group?BSA over LNP with DHA?
Author Response
3. Point-by-point response to Comments and Suggestions for Authors |
Comments 1: |
The manuscript entitled “Cell membrane fatty acids and PIPs modulate the etiology of pancreatic cancer by regulating AKT”, by Carolina Torres et al. as research article, was submitted to Nutrients for possible consideration to be published. PDAC is one worst solid malignancy regarding the outcomes and metabolic dysfunction leading to cachexia. In this study, the authors reported the importance of dietary intervention in treating this disease. The dietary PUFAs like ω3-FA can incorporate into plasma membrane lipids to affect PI3K/AKT signaling and support the emergence of membrane-targeted therapies, as so in this study, abrogate PI3K activity thereby reduce AKT activation and ultimately suppress cell proliferation. Phospholipids in lipid nanoparticles (LNP) impact cell membrane integrity and ultimately cell viability after administration.
Overall, the study was nice but quite many typos and some major issues should be carefully addressed.
1) Regarding the Cell viability assay, I read that each well cell message as authors wrote 5 x 104 cells/ml were used but how about the number of cells per well in the experiments?
Response: Thank you for your comment regarding the Cell Viability Assay. To clarify, for experiments conducted in 96-well plates, cells were seeded at a concentration of 5 x 10⁴ cells/ml, and 100 µl of the cell suspension were added per well, resulting in 5 x 10³ cells per well. For assays performed in larger formats, such as 6-well plates, 2 ml of cell suspension were used per well, and for 10 cm dishes, 10 ml were used per plate.
2) “hours”, “h”, “hr”, “hour”, and “hrs” were used and mixed. Not uniform.
Response: Thank you for pointing out the inconsistency in the usage of time units across the manuscript. We have carefully reviewed and standardized all references to time, ensuring uniformity throughout the text. We now consistently use "h" for hours. We appreciate your attention to detail, which has helped improve the clarity and professionalism of the manuscript.
3) “10% formalin” means what kind concentrations? “1% BSA” means what then? And “5% β-mercaptoethanol”, “0.1% SDS”, “0.05% Tween 20”?
Response: Thank you for highlighting this point. To clarify, the concentrations specified in the manuscript represent the following:
We have now clarified these details in the Materials and Methods section to ensure reproducibility and remove ambiguity. We appreciate your attention to these important details.
4) “40/80μM DHA/LA” was not standard and clear. Response: Thank you for pointing out this issue. We acknowledge that the phrase "40/80 μM DHA/LA" was not clear or standard. This was an error in wording, and we apologize for the confusion caused. To clarify, the intended concentration for DHA and LA used in combination with LNPs was 40 μM.
We have corrected this in the revised manuscript to ensure consistency and clarity as follows: After removal of ethanol under dialysis for 24h the resulting formulation of LNPs dispersed in aqueous buffer was 90% glyceryl monooleate (Sigma-Aldrich) and 10% cholesterol (Sigma-Aldrich) with 40μM of DHA or LA.
Thank you for bringing this to our attention.
5) “Values are expressed as ± SD.” Response: Thank you for pointing this out. To clarify, all values are expressed as mean ± standard deviation (SD) throughout the manuscript, and the number of biological or technical replicates (n) has been indicated in the figure legends and Methods section where applicable.
6) “The significant differences were indicated by *p<0.05; **p<0.005; ***p<0.0005”, and “The specific p<0.05 was considered statistically significant.”
Response: We agree that the second part of the sentence might be repetitive. To avoid redundancy, we have kept only the first sentence, which already clarifies the statistical significance.
7) Results section: how to number the results? Response: Sorry about this. We have corrected the numbering.
8) Figure 2G-H, omega-6 should be more significant that omega-3 for the p-ERK/ERK. But in the figure, both were very similar, and no big difference, compared to SD group.
Response: Thank you for your observation. The results presented in the graphs represent the average of all western blot analyses conducted for the study. However, we acknowledge that the western blot image chosen for Figure 2G-H may not optimally reflect this average due to variability across replicates.
We have reviewed the image and replaced the western blot with one that better reflects the averaged data obtained from multiple samples. This adjustment ensures a more accurate representation of the overall results in the figure. Additionally, we would like to clarify that the bar graph does not represent only the expression levels of pERK but rather the pERK/ERK ratio, which serves as a specific indicator of ERK activation. Thank you for your observation, which has helped us improve the clarity and accuracy of our presentation.
9) Figure 3DEF, 2.5 uM LA showed a sudden increase of viable cells. The cell strain should be labeled in Figure G. The X-axis of Figure H was no correct. Response: Thank you for your observation regarding the increase in cell viability at low doses of LA (2.5 μM). This phenomenon has been noted and is discussed in the text corresponding to these results (page 9). We have highlighted the biological significance of this observation and its potential implications within the context of our study.
We acknowledge the omission of the cell strain label in Figure G. It is indicated in the text, but this is now corrected in the revised version of the figure to ensure clarity and proper identification.
Thank you for pointing out the X-axis in Figure H. Upon review, we confirm that the labeling on the X-axis is correct and consistent with the data presented. If there is a specific issue or ambiguity that might have led to this concern, we would appreciate further clarification to address it effectively.
We are grateful for your thorough feedback and remain committed to ensuring clarity and accuracy throughout the manuscript.
10) Figure 4D, 40 uM LA treatment was underestimated in Fig. 4G. Figure 4B, 5 uM DHA treatment was underestimated in Fig. 4H. Figure 4F, 40 uM LA treatment was underestimated in Fig. 4I. Response: Thank you for your observations regarding Figures 4D and 4G, 4B and 4H, and 4F and 4I. We understand your concern about the apparent underestimation of the effects in the bar graphs compared to the Western blot images.
To clarify, the bar graphs represent the mean values from multiple independent experiments, taking into account the variability inherent in biological replicates. In contrast, the western blot images show a single representative experiment, and it is challenging to select a single blot that perfectly aligns with the averaged data.
Additionally, the bar graphs display the pAKT/AKT ratio as a measure of AKT activation. This ratio accounts for the relative levels of AKT and pAKT, both normalized to the housekeeping protein GAPDH. The comprehensive analysis includes the normalized values of AKT, pAKT, and their ratio, providing a robust measure of activation rather than absolute expression levels.
We appreciate your attention to these details and have ensured that the figure legends and text clarify how the data are presented, helping to avoid any potential misunderstandings. Thank you for your careful review and for allowing us to improve the clarity of our manuscript.
11) In terms of the quantitative analysis, I am not sure the results in Figure 6C and D? Figure 6 E and F? BSA group = LNP group?BSA over LNP with DHA?
|
Response: Thank you for your comment regarding Figure 6C-F. To clarify, Figures 6C and 6E represent results from an immunoprecipitation experiment conducted on cells overexpressing a PIP3 biosensor coupled to GFP. Our hypothesis was that by immunoprecipitating with an anti-GFP antibody and measuring PI3K levels in the precipitate, we could assess the binding of PI3K to PIP3.
In cells treated with DHA, the reduced production of PIP3 by PI3K appears to correlate with a decreased affinity of PI3K for PIP2 in membranes modified by DHA incorporation. This effect is evident in the Western blot of the immunoprecipitation (Figures 6C and 6E) and is further quantified in the bar graphs. Specifically, we observe less PI3K binding in the DHA-treated group compared to LA, when normalized to the input of the immunoprecipitation.
We recognize that the description of these results in the manuscript may require more detail to ensure clarity. We will revise the figure legends and corresponding text to explicitly describe the experimental setup, our hypothesis, and how the results align with the proposed mechanism. Thank you for raising this point, as it allows us to improve the transparency and comprehensibility of our study.
Thank you for your comment regarding Figures 6E and 6F. To clarify, these figures compare the efficiency of DHA delivery mechanisms between two groups:
These comparisons allow us to evaluate how different delivery mechanisms impact the functional effects of DHA on cellular processes. We will ensure that this distinction is clearly described in the figure legends and the corresponding text in the revised manuscript.
Thank you for highlighting this point, which helps us improve the clarity and accuracy of the presentation.
|
